# FedDr+: Stabilizing Dot-regression with Global Feature Distillation for Federated Learning

**Seongyoon Kim**                                            *curisam@kaist.ac.kr*
*Dept. ISysE, KAIST*

**Minchan Jeong**                                            *mcjeong@kaist.ac.kr*
*KAIST AI*

**Sungnyun Kim**                                             *ksn4397@kaist.ac.kr*
*KAIST AI*

**Sungwoo Cho**                                              *peter8526@kaist.ac.kr*
*KAIST AI*

**Sumyeong Ahn**\*                                           *sumyeongahn@kentech.ac.kr*
*KENTECH, Department of Energy Engineering / Energy AI*

**Se-Young Yun**\*                                           *yunseyoung@kaist.ac.kr*
*KAIST AI*

**Reviewed on OpenReview:** *https://openreview.net/forum?id=a6WthNFhL2*

## Abstract

Federated Learning (FL) has emerged as a pivotal framework for the development of effective global models (global FL) or personalized models (personalized FL) across clients with heterogeneous, non-iid data distribution. A key challenge in FL is client drift, where data heterogeneity impedes the aggregation of scattered knowledge. Recent studies have tackled the client drift issue by identifying significant divergence in the last linear (classifier) layer. To mitigate this divergence, strategies such as freezing the classifier weights and aligning the feature extractor accordingly have proven effective. Although the local alignment between classifier and feature extractor has been studied as a crucial factor in FL, we observe that it may lead the model to overemphasize the observed classes and underestimate the unobserved classes within each client. Therefore, our goals are twofold: (1) *improving local alignment* and (2) *maintaining the representation of unseen class samples*, ensuring that the solution seamlessly incorporates knowledge from individual clients, thus enhancing performance in both global and personalized FL. To achieve this, we introduce a novel algorithm named **FedDr+**, which empowers local model alignment using dot-regression loss. **FedDr+** freezes the classifier as a simplex ETF to align the features and improves aggregated global models by employing a feature distillation mechanism to retain information about unseen/missing classes. Our empirical results demonstrate that **FedDr+** not only outperforms methods with a frozen classifier but also surpasses other state-of-the-art approaches, ensuring robust performance across diverse data distributions. The code is available at: *https://github.com/curisam/FedDr_plus*.

---

\* Corresponding authors. This work was done while Sumyeong Ahn was at KAIST.

# 1 Introduction

Federated Learning (FL) (McMahan et al., 2017; He et al., 2020b) is a distributed learning strategy that enables multiple clients to collaboratively train a model while preserving data privacy. The foundational method, FedAvg (McMahan et al., 2017), involves distributing a global model, training local models on each client's private data, and aggregating these models without transmitting raw data. However, a major challenge in FL is data heterogeneity, or *non-iidness*, where differing data distributions across clients lead to *client drift*, hindering the convergence and effectiveness of the global model. Addressing this challenge involves improving two key aspects: *local alignment* and *global knowledge preservation*. Local alignment refers to the cosine similarity between the features extracted by the local model and the classifier's true class vectors, computed on the client's training data, aiming to maximize alignment for improved local training. Global knowledge preservation aims to retain the global model's knowledge of rare or unobserved classes in the client's training data, preventing forgetting during local updates.

While both local alignment and global knowledge preservation are essential, they have generally been studied separately. Global knowledge preservation is crucial because it prevents the model from becoming overly biased toward the data of individual clients, ensuring decisions are based on a broader, shared understanding. This approach enables better generalization across all clients, particularly for unseen classes (Lee et al., 2022; 2024). The challenge of balancing global and local knowledge in FL resembles Catastrophic Forgetting in Continual Learning (CL) (McCloskey & Cohen, 1989), where learning new tasks can cause models to forget previously learned ones. To achieve this, strategies like FedProx (Li et al., 2020), MOON (Li et al., 2021a), and FedNTD (Lee et al., 2022) integrate global model regularization during local training. These methods align local models with the global objective through techniques like proximal terms, contrastive learning, and logit-based regularizers.

On the other direction, to improve the local alignment, recent studies have extensively focused on freezing the last linear layer (called classifier) while updating only the feature extractor. This approach is motivated by the fact that the classifier is most sensitive to data heterogeneity (Luo et al., 2021; Li et al., 2023b; Fan et al., 2024); freezing it ensures that all local models align their features to a consistent classifier across clients. For example, Fed-BABU (Oh et al., 2022) fixes the classifier after initialization, allowing only the feature extractor to adapt. Other methods (Dong et al., 2022; Li et al., 2023b; Fan et al., 2024; Huang et al., 2023; Xiao et al., 2024) further enhance local alignment by modifying the loss function or by using robust initialization techniques, such as the Equiangular Tight Frame (ETF) classifier.

A frozen classifier is also extensively explored in other research areas, such as class imbalance (Yang et al., 2022) and class incremental learning (Yang et al., 2023), with a consistent

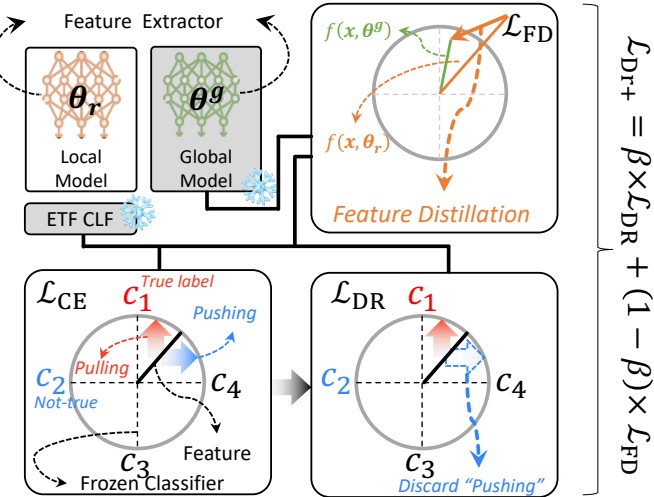

Figure 1: Overview of the proposed method, **FedDr+** trained with $\mathcal{L}_{\text{Dr+}}$. To enhance the local alignment, we employ dot-regression loss $\mathcal{L}_{\text{DR}}$, which discards the pushing term of cross-entropy loss, and propose a feature distillation $\mathcal{L}_{\text{FD}}$ to preserve the knowledge imbued in the global model.

objective similar to aforementioned FL studies—enhancing alignment. Recently, these fields have advanced by introducing and utilizing a novel type of loss, called dot-regression loss $\mathcal{L}_{\text{DR}}$, which aims to achieve alignment rapidly. In summary, $\mathcal{L}_{\text{DR}}$ originates from the decomposition analysis of cross-entropy (CE) loss, which includes *pulling* and *pushing*. As suggested in (Yang et al., 2022), the *pulling* component is a force that attracts features to the target class, whereas the *pushing* component is a force that drives features away from other non-target classes. $\mathcal{L}_{\text{DR}}$ discards the *pushing* component, as it slows down convergence to the desired alignment (refer to Figure 1).

However, our findings indicate that while dot-regression loss enhances *local alignment* as intended, it does not lead to sufficient performance improvement of the aggregated server-side model. The main issue arises from the insufficient *global knowledge preservation* of unobserved classes during local training. Specifically, the focus on improving alignment for classes present in the local training dataset induces significant feature dynamics, which inadvertently disrupt the representation of features associated with unobserved classes. This disruption leads to forgetting and deteriorated alignment for these classes, highlighting the need for better preservation of global knowledge during local training.

We emphasize the importance of addressing both local alignment and global knowledge preservation together. **FedDr+** provides a well-generalized global model by combining dot-regression loss with feature distillation, reducing the distance between feature vectors of local and global models. **FedDr+ FT** extends this by fine-tuning the **FedDr+** global model using the same **FedDr+** loss function, enhancing local alignment for client-specific data. Starting with a well-trained global model is essential for achieving effective personalization while maintaining global generalization (Nguyen et al., 2022; Chen et al., 2023).

**Contributions.** Our main contributions are summarized as follows:

- In high-heterogeneity FL settings, we observe a trade-off in the classifier-freezing setup: dot-regression loss improves local alignment with observed classes but leads to lower global model performance compared to CE loss, due to a significant loss of information on unseen classes, which is critical for the global model.

- To address this, we propose **FedDr+**, which preserves global knowledge through feature distillation while maintaining the advantages of dot-regression loss for local alignment. This contribution focuses on improving global federated learning (GFL).

- We extend **FedDr+** to personalized federated learning (PFL) via **FedDr+ FT**, which fine-tunes the **FedDr+** global model using the **FedDr+** loss function for client-specific data. This highlights the importance of starting with a well-generalized global model for personalization.

- We demonstrate the superiority of our method across various datasets and non-iid settings.

Table 1: Main notations used throughout the paper.

| | |
|---|---|
| **Indices** | |
| $c \in [C]$ | Index for a class |
| $r \in [R]$ | Index for FL round |
| $i \in [N]$ | Index for a client |
| **Dataset** | |
| $D_{\text{train}}^i$ | Training dataset for client $i$ |
| $D_{\text{test}}^i$ | Test dataset for client $i$ |
| $(x,y) \in D_{\text{train,test}}^i \, ; (x,y) \sim \mathcal{D}^i$ | Data on client $i$ sampled from distribution $\mathcal{D}^i$ |
| | ($x$: input data, $y$: class label) |
| $\mathcal{O}^i$ | Dataset consists of observed classes in client $i$ |
| $\mathcal{U}^i$ | Dataset consists of unobserved classes in client $i$ |
| **Parameters** | |
| $\boldsymbol{\theta}$ | Feature extractor weight parameters |
| $\boldsymbol{V} = [v_1, \ldots, v_C] \in \mathbb{R}^{C \times d}$ | Classifier weight parameters (frozen during training) |
| $v_c, c \in [C]$ | $c$-th row vector of $\boldsymbol{V}$ |
| $\boldsymbol{\Theta} = (\boldsymbol{\theta}, \boldsymbol{V})$ | All model parameters |
| $\boldsymbol{\Theta}_r^g = (\boldsymbol{\theta}_r^g, \boldsymbol{V})$ | Aggregated global model parameters at round $r$ |
| $\boldsymbol{\Theta}_r^i = (\boldsymbol{\theta}_r^i, \boldsymbol{V})$ | Trained model parameters on client $i$ at round $r$ |
| **Model Forward** | |
| $p(x; \boldsymbol{\theta}) \in \mathbb{R}^C$ | Softmax probability of input $x$ |
| $p_c(x; \boldsymbol{\theta}), c \in [C]$ | $c$-th element of $p(x; \boldsymbol{\theta})$ |
| $\mathcal{L}_{\text{CE}}(x; \theta) = -\log p_y(x; \boldsymbol{\theta})$ | Cross-entropy loss of input $x$ |
| $f(x; \boldsymbol{\theta}) \in \mathbb{R}^d$ | Feature vector of input $x$ |

## 2 Preliminaries

In this section, we describe the basic settings of Federated Learning (FL), including the `FedAvg` pipeline (McMahan et al., 2017) and the dot-regression loss (Yang et al., 2022), both of which are utilized in our framework. For clarity, we summarize the main notations in Table 1.

### 2.1 Basic Setup of Conventional FedAvg Pipeline

**Basic FL setup.** Let $[N] = \{1, \ldots, N\}$ denote the indices of clients, each with a unique training dataset $D_{\text{train}}^i = \{(x_m, y_m)\}_{m=1}^{|D_{\text{train}}^i|}$, where $(x_m, y_m) \sim \mathcal{D}^i$ for the $i^{\text{th}}$ client, $x_m$ is the input data, and $y_m \in [C]$ is the corresponding label among $C$ classes. Importantly, FL studies predominantly address the scenario where the data distributions are heterogeneous, *i.e.*, $\mathcal{D}^i$ varies across clients. Knowledge distributed among clients is collected over $R$ communication rounds. The general objective of FL is to train a model fit to the aggregated knowledge, $\bigcup_{i \in [N]} \mathcal{D}^i$. This objective can be seen as solving the optimization problem:

$$\min_{\boldsymbol{\Theta} = (\boldsymbol{\theta}, \boldsymbol{V})} \sum_{i \in [N]} \frac{|D_{\text{train}}^i|}{\sum_{j \in [N]} |D_{\text{train}}^j|} \mathbb{E}_{(x,y) \sim \mathcal{D}^i} \Big[ \mathcal{L}(x, y; \boldsymbol{\theta}, \boldsymbol{V}) \Big],$$

where $\mathcal{L}$ is the instance-wise loss function, $\boldsymbol{\theta}$ is the weight parameter for the feature extractor, and $\boldsymbol{V} = [v_1, \ldots, v_C] \in \mathbb{R}^{d \times C}$ is the classifier weight matrix. We use the notation $\boldsymbol{\Theta}$ to denote the entire set of model parameters.

At the beginning of each round $r \in [R]$, the server has access to only a subset of clients $\mathcal{S}_r \subset [N]$ participating in the $r^{\text{th}}$ round. At each round $r$, the server transmits the global model parameters $\boldsymbol{\Theta}_{r-1}^g$ to the participating clients. Each client then updates the parameters with their private data $D_{\text{train}}^i$ and uploads $\boldsymbol{\Theta}_r^i$ to the global server. By incorporating the locally trained weights, the server then updates the global model parameters to $\boldsymbol{\Theta}_r^g$.

**FedAvg pipeline.** Our study follows the conventional FedAvg (McMahan et al., 2017) framework to address the FL problem. FedAvg updates the global model parameters from locally trained parameters by aggregating these local models into $\boldsymbol{\Theta}_r^g = \sum_{i \in S_r} w_r^i \boldsymbol{\Theta}_r^i$, where $w_r^i = |D_{\text{train}}^i| / \sum_{j \in S_r} |D_{\text{train}}^j|$ is the importance weight of the $i^{\text{th}}$ client.

### 2.2 Dot-Regression Loss for Feature Alignment

**Dot-regression loss $\mathcal{L}_{\textbf{DR}}$.** This loss (Yang et al., 2022) facilitates a faster alignment of feature vectors (penultimate layer outputs) $f(x; \boldsymbol{\theta}) \in \mathbb{R}^d$ to the true class direction of $v_y$, reducing the cosine angle as follows:

$$\mathcal{L}_{\text{DR}}(x, y; \boldsymbol{\theta}, \boldsymbol{V}) = \frac{1}{2} \Big( \cos\big(f(x; \boldsymbol{\theta}), v_y\big) - 1 \Big)^2$$

where $\cos(\text{vec}_1, \text{vec}_2)$ denotes the cosine of the angle between two vectors $\angle(\text{vec}_1, \text{vec}_2)$.

The main motivation is that the gradient of the cross-entropy (CE) loss for the feature vector can be decomposed into a *pulling* and *pushing* gradient, and recent work indicates that we can achieve better convergence by removing the pushing effect (Yang et al., 2022; Li & Zhan, 2021). The *pulling* gradient aligns $f(x; \boldsymbol{\theta})$ with $v_y$, while the *pushing* gradient ensures $f(x; \boldsymbol{\theta})$ does not align with $v_c$ for all $c \neq y$ (Appendix B details the exact form of pulling and pushing gradients). Since $\mathcal{L}_{\text{DR}}$ directly attracts features to the true-class classifier, it drops the *pushing* gradient, thereby increasing the convergence speed for maximizing $\cos(f(x; \boldsymbol{\theta}), v_y)$.

**Frozen ETF classifier.** Since $\mathcal{L}_{\text{DR}}$ focuses on aligning feature vectors with the true-class classifier, the classifier is not required to be trained. Instead, we construct the classifier to satisfy the simplex Equiangular Tight Frame (ETF) condition, a constructive way to achieve maximum angular separation between class vectors (Yang et al., 2022; 2023). Concretely, we initialize the classifier weight $\boldsymbol{V}$ as follows and freeze it

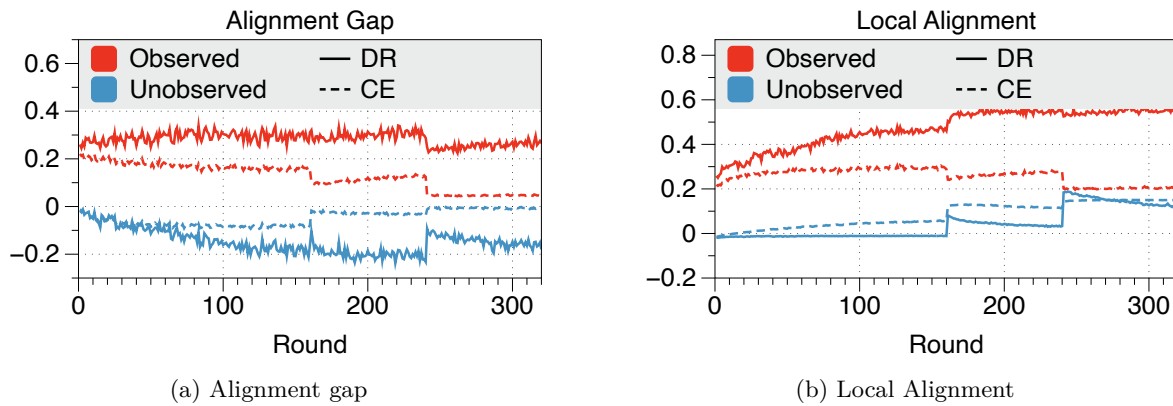

(a) Alignment gap

(b) Local Alignment

Figure 2: Comparison of (a) feature-classifier alignment gap and (b) feature-classifier alignment on the observed and unobserved classes test data for $\boldsymbol{\theta}_r^i$ trained with $\mathcal{L}_{\mathrm{CE}}$ and $\mathcal{L}_{\mathrm{DR}}$.

throughout training:

$$
\boldsymbol{V} \leftarrow \sqrt{\frac{C}{C-1}} \boldsymbol{U} \left( \boldsymbol{I}_C - \frac{1}{C} \mathbf{1}_C \mathbf{1}_C^\top \right),
$$

where $\boldsymbol{U} \in \mathbb{R}^{d \times C}$ is a randomly initialized orthogonal matrix. Note that each $v_i$ in the classifier weight $\boldsymbol{V}$ satisfies $\cos(v_i, v_j) = -\frac{1}{C-1}$ for all $i \neq j \in [C]$[*].

## 3   When Dot-Regression Loss Meets FL

Given our focus on applying $\mathcal{L}_{\mathrm{DR}}$ to FL, we first examine its impact on FL models compared to the CE loss $\mathcal{L}_{\mathrm{CE}}$. In summary, we find that while $\mathcal{L}_{\mathrm{DR}}$ improves alignment-related performance on observed class labels, it faces challenge with unobserved classes[†], which are essential for the generalization objective. To address this issue, we propose **FedDr+**, which integrates $\mathcal{L}_{\mathrm{DR}}$ with a novel feature distillation loss. We then evaluate **FedDr+** by analyzing the effect of feature distillation and compare it with various FL algorithms and regularizers.

**Experimental configuration.**  In this section, we conduct experiments on CIFAR-100 (Krizhevsky et al., 2009) with a shard non-iid setting ($s$=10), where each client contains at most 10 classes. We additionally employ LDA setting ($\alpha$=0.1) in Section 3.4. Refer to Section 4 for more details on the dataset configuration. The model is trained for 320 communication rounds, randomly selecting 10% of clients in each round, and the learning rate is decayed at 160[th] and 240[th] rounds. The experimental configuration for this section is detailed in subsection 4.1.

### 3.1   Impact of $\mathcal{L}_{\mathrm{DR}}$ on Local and Global Models

We analyze the average performance of local and global models trained with $\mathcal{L}_{\mathrm{DR}}$ compared to $\mathcal{L}_{\mathrm{CE}}$, focusing on their ability to generalize. In Figure 2a–2b, we evaluate statistics on two datasets: the observed class set $\mathcal{O}^i$, containing classes in each client's training data $D_{\mathrm{train}}^i$, and the unobserved class set $\mathcal{U}^i$, representing unseen classes. Separately, Figure 3 reports the evaluation on all classes.

First, we examine the amount of change from the given global model to each local model in every communication round (Figure 2a). The alignment gap is denoted by $\cos(f(x; \boldsymbol{\theta}_r^i), v_y) - \cos(f(x; \boldsymbol{\theta}_{r-1}^g), v_y)$. We then evaluate the feature-classifier alignment $\cos(f(x; \boldsymbol{\theta}_r^i), v_y)$ of each local model on the test data (Figure 2b). Finally, we observe the test accuracy of the global model $\boldsymbol{\theta}_r^g$ (Figure 3).

---

[*]This relation for cosines holds if the $v_i$'s are symmetrically distributed such that $\bar{v} = \frac{1}{C} \sum_{i \in [C]} v_i = 0$, and $\cos(v_i, v_j)$ are all the same for $i \neq j$ .

[†]While we use the term "unobserved" in this context, it also applies to "rarely" existing classes.

**Alignment analysis of local models.** As shown in Figure 2a–2b, $\mathcal{L}_{\mathrm{DR}}$ outperforms $\mathcal{L}_{\mathrm{CE}}$ on observed classes in terms of alignment gap and alignment, while $\mathcal{L}_{\mathrm{CE}}$ achieves better results on unobserved classes for these metrics. The improvement on observed classes is attributed to $\mathcal{L}_{\mathrm{DR}}$, which removes the pushing term present in $\mathcal{L}_{\mathrm{CE}}$ and concentrates its pulling effects on these classes. However, this design inherently overlooks unobserved classes, resulting in poorer performance on these classes compared to $\mathcal{L}_{\mathrm{CE}}$.[‡]

**Accuracy result of global models.** Figure 3 shows that in the shard setting ($s = 10$), $\mathcal{L}_{\mathrm{CE}}$ consistently outperforms $\mathcal{L}_{\mathrm{DR}}$. This difference is due to the higher proportion of unobserved in this setting, where each client has access to at most 10 out of 100 classes.

As observed in the alignment analysis, $\mathcal{L}_{\mathrm{DR}}$ performs particularly poorly on the unobserved, with a significantly negative alignment gap and lower alignment compared to models trained with $\mathcal{L}_{\mathrm{CE}}$, which contributes to the lower overall accuracy of the global model when using $\mathcal{L}_{\mathrm{DR}}$.

**Importance of local alignment in observed classes.** While $\mathcal{L}_{\mathrm{DR}}$ struggles with local alignment on unobserved classes, leading to lower global accuracy compared to $\mathcal{L}_{\mathrm{CE}}$, local alignment in observed classes remains critical. At the last learning rate decay round ($240^{\mathrm{th}}$), both methods achieve higher or comparable performance on unobserved classes than in the final round ($320^{\mathrm{th}}$).

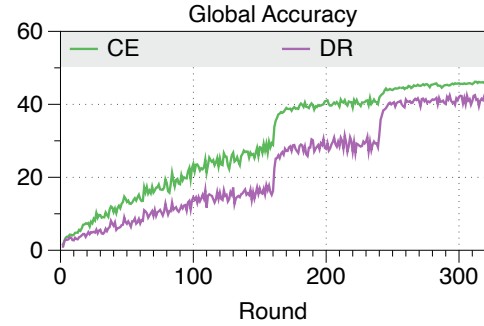

Figure 3: Comparison of the global test accuracy of $\boldsymbol{\theta}_r^g$ on all classes trained using $\mathcal{L}_{\mathrm{CE}}$ and $\mathcal{L}_{\mathrm{DR}}$.

However, the final round shows higher global accuracy due to improved local alignment on observed classes. This underscores the need to preserve the advantages of $\mathcal{L}_{\mathrm{DR}}$ in observed classes while mitigating its degradation on unobserved classes.

### 3.2 FedDr+: $\mathcal{L}_{\mathrm{DR}}$ With Feature Distillation for FL

We propose **FedDr+** to mitigate forgetting unobserved classes while retaining the strengths of dot-regression loss in aligning features of observed classes. Using $\mathcal{L}_{\mathrm{DR}}$ with the frozen classifier $\boldsymbol{V}$, **FedDr+** includes a regularizer that fully distills the global model's feature vectors $f(x; \boldsymbol{\theta}^g) \in \mathbb{R}^d$ to the client features $f(x; \boldsymbol{\theta})$, to enhance generalization across all classes. The proposed loss function $\mathcal{L}_{\mathrm{Dr+}}$, shown in (1), combines $\mathcal{L}_{\mathrm{DR}}$ with a regularizer $\mathcal{L}_{\mathrm{FD}}(x; \boldsymbol{\theta}, \boldsymbol{\theta}^g) = \frac{1}{d}\|f(x; \boldsymbol{\theta}) - f(x; \boldsymbol{\theta}^g)\|_2^2$. Unless specified, we use a scaling parameter $\beta = 0.9$ throughout the paper. The overall pseudocode of **FedDr+** can be found in Appendix A.

$$\mathcal{L}_{\mathrm{Dr+}}(x, y; \boldsymbol{\theta}, \boldsymbol{\theta}^g, \boldsymbol{V}) = \beta \cdot \mathcal{L}_{\mathrm{DR}}(x, y; \boldsymbol{\theta}, \boldsymbol{V}) + (1 - \beta) \cdot \mathcal{L}_{\mathrm{FD}}(x; \boldsymbol{\theta}, \boldsymbol{\theta}^g) \tag{1}$$

**Why feature distillation?** To address data heterogeneity in FL, various distillation methods have been explored, including model parameters (Oh et al., 2022; Li et al., 2020; He et al., 2020a; Li & Wang, 2019), logit-related measurement (Li & Wang, 2019; Lee et al., 2022; Itahara et al., 2021; Ye et al., 2024; Lin et al., 2020; Chen et al., 2019; Qian et al., 2022), and co-distillation (Chen et al., 2024; Cho et al., 2023). In contrast, we utilize the *feature* distillation (Heo et al., 2019) technique because the feature directly concerns alignment. On the other hand, logits lose information from features when projected onto a frozen ETF classifier (Heo et al., 2019; Li et al., 2017; 2023a; Ben-Baruch et al., 2022). By distilling features, we leverage the global, differentiated knowledge for each data input $x$. This approach aims to minimize drift towards observed classes, and hence, we expect it to enhance overall generalization.

### 3.3 Effect of Feature Distillation

Our findings from Section 3.1 indicate that $\mathcal{L}_{\mathrm{DR}}$ is unsuitable for the heterogeneous FL environment. This is primarily because there is a notable gap in how features align with the fixed classifier between $\mathcal{O}^i$ and

---

[‡]A theoretical justification for why $\mathcal{L}_{\mathrm{DR}}$ struggles with unobserved classes is provided in Appendix C, where we analyze feature gradients under the NTK framework.

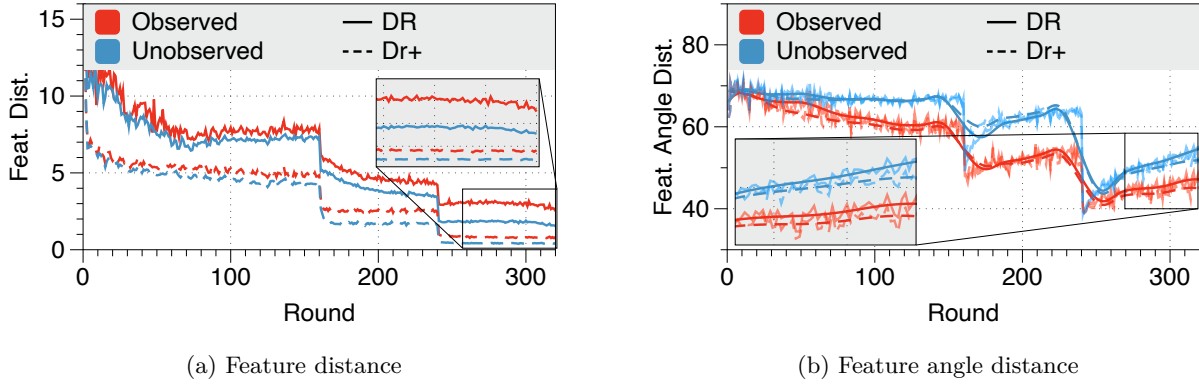

(a) Feature distance

(b) Feature angle distance

Figure 4: We present (a) feature distance and (b) feature angle distance from $\boldsymbol{\theta}_{r-1}^g$ to $\boldsymbol{\theta}_r^i$ for observed and unobserved classes by training with $\mathcal{L}_{\mathrm{DR}}$ and $\mathcal{L}_{\mathrm{Dr+}}$.

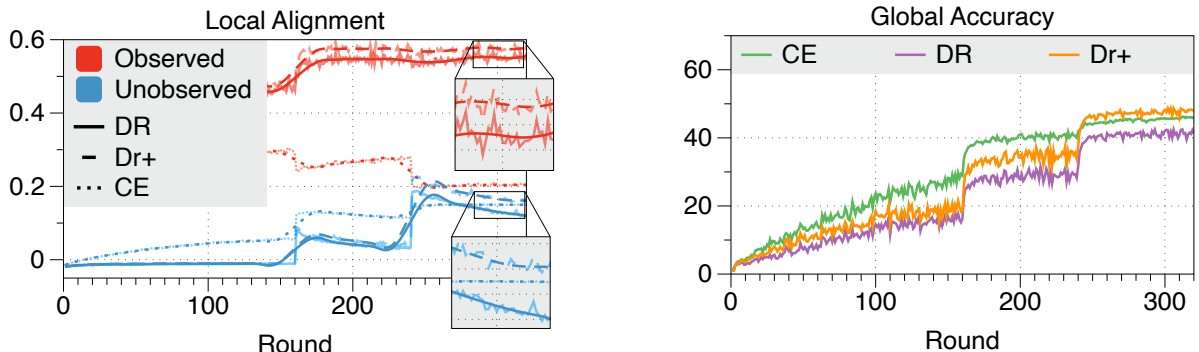

Figure 5: Comparison of feature-classifier alignment on the observed and unobserved classes test data for $\boldsymbol{\theta}_r^i$ trained with $\mathcal{L}_{\mathrm{CE}}, \mathcal{L}_{\mathrm{DR}}$ and $\mathcal{L}_{\mathrm{Dr+}}$.

Figure 6: Comparison of the global test accuracy of $\boldsymbol{\theta}_r^g$ on all classes trained using $\mathcal{L}_{\mathrm{CE}}, \mathcal{L}_{\mathrm{DR}}$ and $\mathcal{L}_{\mathrm{Dr+}}$.

$\mathcal{U}^i$. To assess the effect of feature distillation ($\mathcal{L}_{\mathrm{FD}}$), which imposes a constraint on the feature distance $\|f(x; \boldsymbol{\theta}_r^i) - f(x; \boldsymbol{\theta}_{r-1}^g)\|_2$ for $x \in \mathcal{O}^i$, we measure this distance for both $\mathcal{O}^i$ and $\mathcal{U}^i$ from the models trained with $\mathcal{L}_{\mathrm{DR}}$ and $\mathcal{L}_{\mathrm{Dr+}}$. We additionally analyze the angle distance, $\angle(f(x; \boldsymbol{\theta}_r^i), f(x; \boldsymbol{\theta}_{r-1}^g))$, as it impacts feature-classifier alignment. These values are averaged over the selected client set $\mathcal{S}_r$.

**Feature distillation stabilizes the feature dynamics.** By adding $\mathcal{L}_{\mathrm{FD}}$, as revealed in Figure 4a, the local model trained with $\mathcal{L}_{\mathrm{Dr+}}$ shows a reduction in feature distance for observed classes, compared to the model trained with $\mathcal{L}_{\mathrm{DR}}$. This reduction happens even for unobserved classes. As demonstrated in Figure 4b, the overall decrease in feature distance leads to a reduction in feature angle distance for both class sets. In both local models trained with $\mathcal{L}_{\mathrm{DR}}$ and $\mathcal{L}_{\mathrm{Dr+}}$, there is a trend where the angle distance is significantly larger for $\mathcal{U}^i$ than for $\mathcal{O}^i$ (Figure 4b). This large angle distance of $\mathcal{U}^i$ leads to the degradation of the feature-classifier alignment.

**Feature distillation enhances local alignment and global accuracy.** As in Figure 5–6, our proposed algorithm, *i.e.*, $\mathcal{L}_{\mathrm{Dr+}}$, improves feature-classifier alignment for both $\mathcal{O}^i$ and $\mathcal{U}^i$, along with enhanced global accuracy compared to $\mathcal{L}_{\mathrm{DR}}$. We attribute this improvement to the enhanced knowledge of the global model which is preserved by preventing the forgetting of previously trained knowledge. During the final convergence phase ($240^{\mathrm{th}}$-$320^{\mathrm{th}}$), $\mathcal{L}_{\mathrm{Dr+}}$ achieves better feature-classifier alignment even on unobserved classes compared to $\mathcal{L}_{\mathrm{CE}}$. As a result, at the final round ($320^{\mathrm{th}}$), $\mathcal{L}_{\mathrm{Dr+}}$ demonstrates the best local alignment across all classes, surpassing both $\mathcal{L}_{\mathrm{CE}}$ and $\mathcal{L}_{\mathrm{DR}}$, contributing to its global model's superior performance. Even though the

Table 2: Synergy of various FL algorithms and regularizers. Baseline indicates training FL models without a regularizer. FD denotes feature distillation, which is the regularizer we use in **FedDr+**.

| Algorithm | Sharding ($s = 10$) | | | | | | | LDA ($\alpha = 0.1$) | | | | | | |
|---|---|---|---|---|---|---|---|---|---|---|---|---|---|---|
| | Baseline | +Prox | +KD | +NTD | +LD | +MOON | +FD | Baseline | +Prox | +KD | +NTD | +LD | +MOON | +FD |
| FedAvg | 37.22 | 36.87 | 36.25 | 37.71 | 37.17 | 37.43 | 37.82 | 42.52 | 43.22 | 44.21 | 43.39 | 43.43 | 44.79 | 43.76 |
| FedBABU | 46.20 | 46.03 | 46.37 | 47.22 | 46.71 | 46.49 | 46.95 | 47.37 | 46.62 | 47.60 | 46.48 | 45.78 | 46.27 | 46.49 |
| SphereFed | 43.90 | 41.96 | 44.94 | 43.47 | 43.95 | 43.13 | 45.21 | 46.98 | 43.77 | 47.76 | 47.25 | 47.01 | 46.81 | 49.74 |
| FedETF | 32.42 | 31.87 | 32.76 | 32.65 | 32.25 | 34.30 | 32.77 | 46.27 | 45.71 | 46.67 | 46.16 | 45.91 | 45.98 | 46.47 |
| FedGELA | 29.17 | 28.69 | 29.11 | 28.84 | 29.36 | 28.80 | 30.33 | 27.11 | 29.03 | 28.45 | 29.62 | 29.41 | 28.09 | 29.75 |
| Dot-Regression | 42.52 | 41.95 | 47.45 | 48.32 | 47.52 | 44.72 | **48.69** | 42.72 | 46.35 | 49.47 | 50.36 | 49.28 | 50.36 | **50.86** |

proposed regularizer demonstrates a reasonable regularizing effect, one question remains: "*Is it superior to other previously used regularizers?*"

### 3.4 Synergistic Effect with Different Types of FL Algorithms and Regularizers

We answer the above question by evaluating the synergy effect of various FL algorithms by maintaining their original training loss and incorporating specific regularizers, following the approach suggested in Equation 1. To address the issue of differing loss scales between the baseline FL algorithms and the regularizers, we thoroughly tune the coefficient $\beta$ within the range $\{0.1, 0.3, 0.5, 0.7, 0.9, 0.99, 0.999, 0.9999\}$, and report the resulting performance in Table 2. The selected $\beta$ values are detailed in Appendix E.

**FL algorithms and regularizers.** We evaluate several baseline FL algorithms, including dot-regression. The baseline also include FedAvg (McMahan et al., 2017)—using an unfrozen classifier, FedBABU (Oh et al., 2022)—extending FedAvg by freezing the classifier during local training, and SphereFed(Dong et al., 2022)—enhancing feature-classifier alignment by using MSE loss between one-hot encoded labels and cosine similarity based logits. We also considered FedGELA and FedETF—both applying client-specific adaptive loss functions tailored to data distribution.

Alongside the FD regularizer, we evaluate a range of regularizers, including Prox (Li et al., 2020)—constraining the distance between local and global model parameters; MOON (Li et al., 2021a)—minimizing the angle distance between feature vectors of global and local models through contrastive learning; and several logit-based regularizers—KD (Hinton et al., 2015), NTD (Lee et al., 2022; Zhao et al., 2022), and LD (Kim et al., 2021)—keeping logit-related measurements of local models closely aligned with the global model. Specifically, KD applies softened softmax probability from the logit vector, NTD does the same but excludes the true class dimension, and LD distills the entire logit vector.

**FedDr+ shows best synergy.** Table 2 shows that **FedDr+** (dot-regression + FD) achieves the strongest performance among the tested combinations. FD performs exceptionally well when combined with dot-regression, SphereFed, and FedBABU. These baselines freeze the classifier while not using client-specific adaptive loss. Among them, the synergy is most effective in the order of dot-regression, SphereFed, and FedBABU—reflecting how each optimizes feature-classifier alignment well. FD outperforms other regularizers by effectively stabilizing feature dynamics for observed classes, which helps mitigate the misalignment and performance degradation for unobserved classes. MOON,on the other hand, prioritizes the cosine similarity between feature vectors from local and global models but fails to adequately control feature norm dynamics. Prox applies uniform regularization that is independent of specific data instances, leading to less refined control over feature dynamics. Logit-based regularizers lose effectiveness due to information loss when features are projected onto the classifier, as they focus on mitigating the dynamics of the less informative projected vectors rather than the richer original feature vectors.

For baselines like FedGELA and FedETF, which apply client-specific loss functions, none of the regularizers, including FD, lead to consistently lead to significant performance, particularly in the sharding setting. In non-frozen settings, FedAvg, the FD regularizer does not offer a significant performance boost. Under LDA, FedAvg combined with MOON outperforms FedAvg with FD, consistent with the claim of MOON (Li et al., 2021a).

# 4 Experiments and Results

In this section, we first present the experimental results of **FedDr+** in the context of global federated learning (GFL). We then analyze the elapsed time and conduct a sensitivity analysis for GFL, investigating the effects of varying local epochs, client sampling ratios, and different $\beta$ values on the performance of **FedDr+**. Finally, we propose **FedDr+** FT, which fine-tunes the **FedDr+** GFL model with $\mathcal{L}_{\text{Dr+}}$, and report the results comparing it with existing PFL methods.

## 4.1 Experimental Setup

**Dataset and models.** To simulate a realistic FL scenario involving 100 clients, we conduct extensive studies on three widely used datasets: CIFAR-10 (Krizhevsky et al., 2009), CIFAR-100 (Krizhevsky et al., 2009) and ImageNet-100 (Deng et al., 2009). We use VGG11 (Simonyan & Zisserman, 2014) for CIFAR-10, MobileNet (Howard et al., 2017) for CIFAR-100, and ResNet-18 (He et al., 2016) for ImageNet-100. The training data is distributed among the 100 clients using sharding and the LDA (Latent Dirichlet Allocation) partition strategies.

Following the convention, sharding distributes the data into non-overlapping shards of equal size, each shard encompassing $\frac{|D_{\text{train}}|}{100 \times s}$ and $\frac{|D_{\text{test}}|}{100 \times s}$ samples per class, where $s$ denotes the number of shards per client. On the other hand, LDA involves sampling a probability vector from Dirichlet distribution, $p_c = (p_{c,1}, p_{c,2}, \cdots, p_{c,100}) \sim \text{Dir}(\alpha)$, and allocating a proportion $p_{c,k}$ of instances of class $c \in [C]$ to each client $k \in [100]$. Smaller values of $s$ and $\alpha$ increase the level of data heterogeneity. For CIFAR-10 and CIFAR-100, we explore a range of $s$ and $\alpha$ values to assess the impact of different data heterogeneity levels. For ImageNet-100, we focus on experiments with $s = 20$ and $\alpha = 0.1$.

**Implementation details.** In each round of communication, a random 10% of clients are selected to participate in the training process. The total number of communication rounds is set to 320. The initial learning rate and the number of local epochs for CIFAR-10, CIFAR-100, and ImageNet-100 are determined through grid searches, with the detailed process and results provided in Appendix D. The learning rate $\eta$ is decayed by a factor of 0.1 at the 160th and 240th communication rounds.

## 4.2 Global Federated Learning Results

We compare **FedDr+** with a range of GFL algorithms, considering both non-freezing and freezing classifier approaches. Among non-freezing classifiers, **FedDr+** competes with FedAvg (McMahan et al., 2017), Fed-Prox (Li et al., 2020), SCAFFOLD (Karimireddy et al., 2020), MOON (Li et al., 2021a), FedNTD (Lee et al., 2022), FedExP (Jhunjhunwala et al., 2023), and FedSOL (Lee et al., 2024). **FedDr+** is also evaluated against freezing classifier algorithms such as FedBABU (Oh et al., 2022), SphereFed (Dong et al., 2022), FedETF (Li et al., 2023b), and FedGELA (Fan et al., 2024). Among the baseline algorithms, SCAFFOLD incurs a communication cost two times higher per round, denoted as (×2). Our experiments encompass heterogeneous settings involving sharding and LDA non-IID environments.

Table 3 summarizes the accuracy comparison between various GFL methods proposed in recent literature and FedAvg under various conditions. While specific methods demonstrated effectiveness in particular scenarios, some of these underperformed relative to the robustness of FedAvg. For example, SCAFFOLD shown strong performance in the less heterogeneous sharding setting on CIFAR-10; however, it failed in model training under the highly heterogeneous LDA condition with $\alpha = 0.1$. Notably, **FedDr+** consistently outperformed all baselines in highly heterogeneous settings, achieving a 3.15% improvement in CIFAR-100 LDA with $\alpha = 0.05$ and 3.17% in ImageNet-100 LDA.

## 4.3 Elapsed Time Results

We compare **FedDr+** with various GFL algorithms for the elapsed time per communication round on CIFAR-100 ($s$=10). As shown in Table 4, incorporating a global model during the local update generally results in higher computation costs, leading to longer elapsed times compared to updates without a global model.

Table 3: Accuracy comparison in the GFL setting. The entries are based on results obtained from three different seeds, indicating the mean and standard deviation of the accuracy of the global model, represented as $X_{\pm Y}$. The best performance in each case is highlighted in **bold**.

| | | | NIID Partition Strategy: Sharding | | | | | |
|---|---|---|---|---|---|---|---|---|
| | CIFAR-100 | | | | CIFAR-10 | | | ImageNet-100 |
| | $s=10$ | $s=20$ | $s=50$ | $s=100$ | $s=2$ | $s=5$ | $s=10$ | |
| FedAvg | $36.63_{\pm 0.22}$ | $42.25_{\pm 1.42}$ | $45.57_{\pm 0.22}$ | $48.20_{\pm 1.36}$ | $72.08_{\pm 0.67}$ | $81.53_{\pm 0.35}$ | $82.38_{\pm 0.40}$ | $67.78_{\pm 0.41}$ |
| FedProx | $37.07_{\pm 0.21}$ | $42.35_{\pm 0.83}$ | $45.18_{\pm 1.07}$ | $47.78_{\pm 0.79}$ | $71.92_{\pm 0.51}$ | $81.29_{\pm 0.40}$ | $82.45_{\pm 0.35}$ | $67.81_{\pm 0.65}$ |
| SCAFFOLD($\times$2) | $46.08_{\pm 0.37}$ | $48.15_{\pm 1.21}$ | $49.31_{\pm 0.62}$ | $50.73_{\pm 0.42}$ | $75.49_{\pm 0.42}$ | $\mathbf{84.14}_{\pm 0.13}$ | $\mathbf{85.11}_{\pm 0.29}$ | $70.47_{\pm 0.46}$ |
| MOON | $36.95_{\pm 0.37}$ | $43.05_{\pm 0.27}$ | $43.95_{\pm 0.12}$ | $46.92_{\pm 0.08}$ | $67.55_{\pm 1.16}$ | $80.90_{\pm 0.26}$ | $82.62_{\pm 0.31}$ | $68.19_{\pm 0.32}$ |
| FedNTD | $34.05_{\pm 1.19}$ | $41.78_{\pm 0.31}$ | $46.42_{\pm 0.63}$ | $47.17_{\pm 0.32}$ | $72.21_{\pm 0.59}$ | $69.96_{\pm 17.10}$ | $81.99_{\pm 0.42}$ | $67.51_{\pm 0.25}$ |
| FedExP | $36.85_{\pm 1.11}$ | $42.49_{\pm 1.22}$ | $45.07_{\pm 0.92}$ | $48.09_{\pm 1.00}$ | $72.31_{\pm 0.60}$ | $81.41_{\pm 0.19}$ | $82.47_{\pm 0.16}$ | $63.34_{\pm 0.51}$ |
| FedSOL | $32.18_{\pm 0.18}$ | $41.54_{\pm 1.03}$ | $47.42_{\pm 0.76}$ | $47.70_{\pm 1.13}$ | $54.93_{\pm 3.52}$ | $75.73_{\pm 0.10}$ | $77.00_{\pm 0.41}$ | $66.61_{\pm 1.17}$ |
| FedBABU | $45.97_{\pm 0.48}$ | $45.53_{\pm 0.79}$ | $46.52_{\pm 0.51}$ | $46.02_{\pm 0.28}$ | $71.99_{\pm 0.52}$ | $81.07_{\pm 0.60}$ | $82.32_{\pm 0.06}$ | $68.82_{\pm 0.46}$ |
| SphereFed | $42.71_{\pm 0.65}$ | $48.63_{\pm 0.90}$ | $\mathbf{52.16}_{\pm 0.22}$ | $\mathbf{53.41}_{\pm 0.19}$ | $\mathbf{76.33}_{\pm 0.33}$ | $83.67_{\pm 0.18}$ | $84.36_{\pm 0.30}$ | $69.71_{\pm 0.39}$ |
| FedETF | $31.37_{\pm 0.72}$ | $42.22_{\pm 0.77}$ | $47.47_{\pm 0.67}$ | $49.00_{\pm 0.74}$ | $67.81_{\pm 0.94}$ | $80.78_{\pm 0.68}$ | $82.60_{\pm 0.46}$ | $70.81_{\pm 0.28}$ |
| FedGELA | $27.95_{\pm 0.81}$ | $38.63_{\pm 0.66}$ | $44.67_{\pm 0.51}$ | $47.95_{\pm 0.85}$ | $63.77_{\pm 2.34}$ | $79.05_{\pm 0.14}$ | $81.56_{\pm 0.09}$ | $67.08_{\pm 0.18}$ |
| **FedDr+ (Ours)** | $\mathbf{48.21}_{\pm 0.56}$ | $\mathbf{50.77}_{\pm 0.14}$ | $52.15_{\pm 0.03}$ | $52.41_{\pm 0.81}$ | $76.57_{\pm 0.51}$ | $83.22_{\pm 0.34}$ | $84.14_{\pm 0.27}$ | $\mathbf{71.47}_{\pm 0.45}$ |

| | | | NIID Partition Strategy: LDA | | | | | |
|---|---|---|---|---|---|---|---|---|
| | CIFAR-100 | | | | CIFAR-10 | | | ImageNet-100 |
| | $\alpha=0.05$ | $\alpha=0.1$ | $\alpha=0.2$ | $\alpha=0.3$ | $\alpha=0.1$ | $\alpha=0.2$ | $\alpha=0.3$ | |
| FedAvg | $35.58_{\pm 1.35}$ | $42.10_{\pm 0.60}$ | $44.78_{\pm 0.72}$ | $45.73_{\pm 0.88}$ | $68.71_{\pm 1.82}$ | $77.75_{\pm 0.26}$ | $80.76_{\pm 0.51}$ | $65.11_{\pm 0.25}$ |
| FedProx | $37.07_{\pm 0.21}$ | $42.35_{\pm 0.83}$ | $45.18_{\pm 1.06}$ | $48.18_{\pm 0.51}$ | $69.00_{\pm 2.27}$ | $77.81_{\pm 0.24}$ | $80.55_{\pm 0.19}$ | $40.48_{\pm 1.28}$ |
| SCAFFOLD($\times$2) | $40.54_{\pm 0.48}$ | $46.14_{\pm 0.70}$ | $47.98_{\pm 0.93}$ | $48.06_{\pm 1.08}$ | *(Failed)* | $80.15_{\pm 0.29}$ | $\mathbf{82.63}_{\pm 0.23}$ | $66.84_{\pm 0.77}$ |
| MOON | $23.97_{\pm 1.15}$ | $30.86_{\pm 0.21}$ | $33.60_{\pm 0.62}$ | $35.54_{\pm 0.45}$ | $66.44_{\pm 3.28}$ | $77.36_{\pm 0.08}$ | $80.15_{\pm 0.22}$ | $41.25_{\pm 0.60}$ |
| FedNTD | $31.78_{\pm 3.14}$ | $40.41_{\pm 0.96}$ | $43.10_{\pm 2.03}$ | $43.04_{\pm 0.82}$ | $70.22_{\pm 0.40}$ | $77.16_{\pm 0.20}$ | $79.50_{\pm 0.56}$ | $64.87_{\pm 0.20}$ |
| FedExP | $34.39_{\pm 1.77}$ | $40.85_{\pm 1.32}$ | $44.47_{\pm 0.28}$ | $45.44_{\pm 0.14}$ | $70.14_{\pm 0.53}$ | $78.09_{\pm 0.21}$ | $80.40_{\pm 0.54}$ | $59.40_{\pm 0.36}$ |
| FedSOL | $34.49_{\pm 0.80}$ | $41.19_{\pm 0.30}$ | $43.55_{\pm 1.51}$ | $44.85_{\pm 0.54}$ | $59.51_{\pm 1.77}$ | $67.55_{\pm 0.41}$ | $70.96_{\pm 0.32}$ | $62.70_{\pm 0.89}$ |
| FedBABU | $41.97_{\pm 1.01}$ | $45.77_{\pm 0.28}$ | $44.28_{\pm 0.45}$ | $44.80_{\pm 0.63}$ | $65.15_{\pm 3.66}$ | $77.03_{\pm 0.25}$ | $79.91_{\pm 0.13}$ | $66.54_{\pm 0.30}$ |
| SphereFed | $39.56_{\pm 0.48}$ | $46.54_{\pm 0.58}$ | $\mathbf{49.41}_{\pm 0.78}$ | $49.22_{\pm 0.86}$ | $67.49_{\pm 3.49}$ | $80.05_{\pm 0.40}$ | $82.62_{\pm 0.66}$ | $67.03_{\pm 0.30}$ |
| FedETF | $40.71_{\pm 0.90}$ | $45.63_{\pm 0.33}$ | $46.28_{\pm 1.05}$ | $46.69_{\pm 0.87}$ | $70.75_{\pm 0.36}$ | $77.86_{\pm 0.46}$ | $79.95_{\pm 0.34}$ | $68.98_{\pm 0.21}$ |
| FedGELA | $16.72_{\pm 1.91}$ | $27.12_{\pm 1.58}$ | $33.68_{\pm 0.19}$ | $36.17_{\pm 0.26}$ | $50.69_{\pm 7.55}$ | $66.04_{\pm 14.87}$ | $77.89_{\pm 0.97}$ | $55.57_{\pm 0.42}$ |
| **FedDr+ (Ours)** | $\mathbf{45.12}_{\pm 1.00}$ | $\mathbf{49.48}_{\pm 0.50}$ | $50.67_{\pm 0.88}$ | $\mathbf{51.15}_{\pm 0.65}$ | $\mathbf{72.07}_{\pm 2.26}$ | $\mathbf{80.90}_{\pm 0.02}$ | $82.42_{\pm 0.10}$ | $\mathbf{70.20}_{\pm 0.09}$ |

Table 4: Elapsed time per round (in seconds) for various GFL algorithms.

| | Local update without global model | | | | | | Local update with global model | | | | | |
|---|---|---|---|---|---|---|---|---|---|---|---|---|
| | FedAvg | FedBABU | FedETF | SphereFed | FedGELA | FedExP | FedProx | SCAFFOLD | FedNTD | FedSOL | MOON | **FedDr+** (Ours) |
| Elapsed time | 20.4 | 20.5 | 20.7 | 20.4 | 20.5 | 20.1 | 22.6 | 21.4 | 21.2 | 27.2 | 56.7 | 24.9 |

**FedDr+** exhibits a slightly longer elapsed time than most other algorithms, yet still requires less time than FedSOL and MOON.

### 4.4 Sensitivity Analysis

We explore the impact of varying client sampling ratio and local epochs on performance, as well as the effect of different $\beta$ values in **FedDr+**, as detailed in Figure 7. All experiments are conducted on MobileNet using the CIFAR-100 dataset with LDA ($\alpha=0.1$).

**Effect of client sampling ratio and local epochs.** We evaluate the sensitivity of hyperparameters in **FedDr+** by comparing it to baselines under varying client sampling ratio and local epochs, starting from the default setting of client sampling ratio of 0.1 and local epoch of 3. Compared to FedAvg (without classifier freezing), FedBABU and SphereFed (all with classifier freezing) generally show performance improvements with increasing fraction ratios, but **FedDr+** consistently outperforms the baselines. The number of local epochs is crucial in FL; too few epochs result in underfitting, while too many cause client drift, degrading

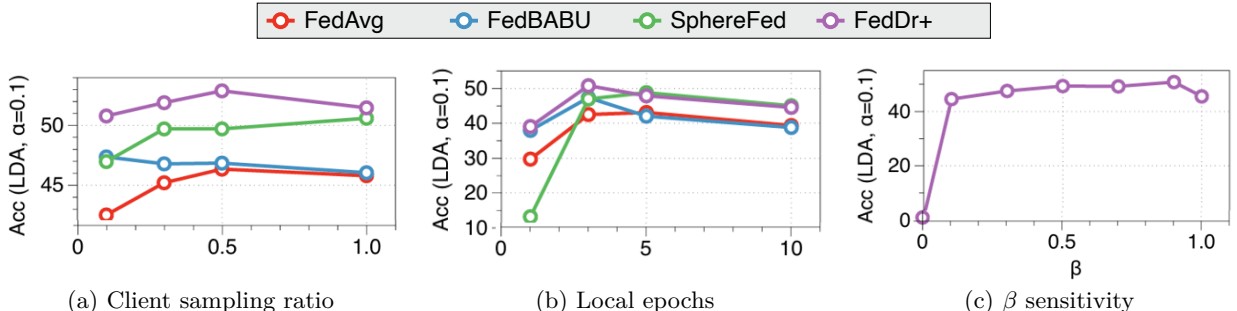

(a) Client sampling ratio    (b) Local epochs    (c) $\beta$ sensitivity

Figure 7: Performance of baselines and **FedDr+** on CIFAR-100 ($\alpha$=0.1) with various analyses: (a) client sampling ratio, (b) the number of local epochs, and (c) sensitivity to $\beta$.

Table 5: PFL accuracy comparison with MobileNet on CIFAR-100. Results are reported in the format $X_{\pm Y}$, representing the mean and standard deviation of the average personalized accuracies across all clients, computed over five seeds. The best performance in each case is highlighted in **bold**.

| Algorithm | $s$=10 | $s$=20 | $s$=100 | $\alpha$=0.05 | $\alpha$=0.1 | $\alpha$=0.3 |
|---|---|---|---|---|---|---|
| Local only ($\mathcal{L}_{\mathrm{CE}}$) | $58.42_{\pm\ 0.22}$ | $42.37_{\pm\ 0.29}$ | $19.02_{\pm\ 0.34}$ | $55.71_{\pm\ 0.19}$ | $44.02_{\pm\ 0.39}$ | $27.98_{\pm\ 0.08}$ |
| Local only ($\mathcal{L}_{\mathrm{CE}}$+ETF) | $58.05_{\pm\ 0.25}$ | $41.72_{\pm\ 0.26}$ | $19.06_{\pm\ 0.18}$ | $55.41_{\pm\ 0.21}$ | $43.56_{\pm\ 0.31}$ | $27.69_{\pm\ 0.17}$ |
| Local only ($\mathcal{L}_{\mathrm{DR}}$) | $61.05_{\pm\ 0.37}$ | $44.28_{\pm\ 0.21}$ | $21.06_{\pm\ 0.21}$ | $58.56_{\pm\ 0.14}$ | $47.05_{\pm\ 0.13}$ | $31.16_{\pm\ 0.15}$ |
| FedPer | $70.62_{\pm\ 0.71}$ | $55.65_{\pm\ 1.35}$ | $25.57_{\pm\ 0.59}$ | $63.35_{\pm\ 1.96}$ | $51.90_{\pm\ 2.13}$ | $35.84_{\pm\ 2.16}$ |
| Per-FedAvg | $31.71_{\pm\ 1.08}$ | $38.64_{\pm\ 0.40}$ | $45.71_{\pm\ 0.81}$ | $28.85_{\pm\ 0.27}$ | $36.00_{\pm\ 0.42}$ | $42.41_{\pm\ 0.32}$ |
| FedRep | $62.59_{\pm\ 0.30}$ | $51.18_{\pm\ 1.00}$ | $26.51_{\pm\ 0.27}$ | $57.73_{\pm\ 0.41}$ | $49.59_{\pm\ 0.40}$ | $36.22_{\pm\ 0.86}$ |
| Ditto | $38.39_{\pm\ 0.54}$ | $42.16_{\pm\ 1.14}$ | $44.04_{\pm\ 0.81}$ | $34.86_{\pm\ 1.18}$ | $38.67_{\pm\ 1.30}$ | $42.05_{\pm\ 0.58}$ |
| FedAvg-FT | $70.20_{\pm\ 0.54}$ | $56.26_{\pm\ 0.51}$ | $48.67_{\pm\ 0.99}$ | $61.08_{\pm\ 1.86}$ | $56.34_{\pm\ 1.18}$ | $49.74_{\pm\ 1.08}$ |
| FedBABU-FT | $80.73_{\pm\ 0.65}$ | $71.02_{\pm\ 0.34}$ | $51.70_{\pm\ 0.21}$ | $76.12_{\pm\ 0.55}$ | $69.94_{\pm\ 0.34}$ | $57.40_{\pm\ 1.50}$ |
| SphereFed-FT | $81.34_{\pm\ 0.64}$ | $72.22_{\pm\ 0.56}$ | $56.58_{\pm\ 0.89}$ | $74.49_{\pm\ 0.86}$ | $69.39_{\pm\ 1.04}$ | $59.51_{\pm\ 1.03}$ |
| FedETF-FT | $53.32_{\pm\ 0.60}$ | $53.05_{\pm\ 0.49}$ | $49.74_{\pm\ 0.85}$ | $52.31_{\pm\ 0.40}$ | $53.70_{\pm\ 0.35}$ | $50.80_{\pm\ 0.65}$ |
| FedGELA-FT | $75.75_{\pm\ 0.57}$ | $68.96_{\pm\ 0.37}$ | $52.23_{\pm\ 0.59}$ | $58.26_{\pm\ 5.78}$ | $60.12_{\pm\ 0.71}$ | $53.09_{\pm\ 0.82}$ |
| **FedDr+ FT (ours)** | $\mathbf{83.08_{\pm\ 0.27}}$ | $\mathbf{74.80_{\pm\ 0.66}}$ | $\mathbf{56.56_{\pm\ 1.04}}$ | $\mathbf{78.40_{\pm\ 0.40}}$ | $\mathbf{73.23_{\pm\ 0.89}}$ | $\mathbf{62.22_{\pm\ 0.86}}$ |

global model performance. The default setting of local epochs 3 is optimal for all baselines, with **FedDr+** achieving the best performance. Although performance generally declines when deviating from this peak point, **FedDr+** remains the best or highly competitive.

**Weight ratio $\beta$ analysis.** We analyze the effect of scaling parameter in **FedDr+** by varying $\beta$ while keeping other hyperparameters constant. The performance is evaluated for $\beta \in \{0, 0.1, 0.3, 0.5, 0.7, 0.9, 1.0\}$. When $\beta = 0$, only feature distillation is applied, and when $\beta = 1$, only dot-regression is used. $\beta \in \{0, 1\}$ are generally less effective, whereas $\beta \in \{0.3, 0.5, 0.7, 0.9\}$ show consistently good performance, indicating a balanced approach is beneficial.

## 4.5 Personalized Federated Learning Results

We introduce **FedDr+ FT**, inspired by prior work (Oh et al., 2022; Dong et al., 2022; Li et al., 2023b; Kim et al., 2023; Fan et al., 2024), which enhances personalization by leveraging local data to fine-tune the global federated learning (GFL) model. We fine-tune the **FedDr+** GFL model using $\mathcal{L}_{\mathrm{Dr+}}$ to create **FedDr+ FT**, *i.e.,* 2-step approach. The overall pseudocode of **FedDr+ FT** can be found in Appendix A. For a comprehensive analysis, we compare **FedDr+ FT** with existing personalized federated learning (PFL) methods, including 1-step approaches, *i.e.,* creating PFL models from scratch, such as FedPer (Arivazhagan et al., 2019), Per-FedAvg (Fallah et al., 2020), FedRep (Collins et al., 2021), and Ditto (Li et al., 2021b), as well as 2-step methods such as FedAVG-FT, FedBABU-FT (Oh et al., 2022), SphereFed-FT (Dong et al., 2022), FedETF-FT (Li et al., 2023b), and FedGELA-FT (Fan et al., 2024). Additionally, we compare these methods with various simple local models that have not undergone federated learning: (1) Local only ($\mathcal{L}_{\mathrm{CE}}$),

trained with $\mathcal{L}_{CE}$, (2) Local only ($\mathcal{L}_{CE}$ + ETF), trained with $\mathcal{L}_{CE}$ and initializing the classifier with an ETF classifier, and (3) Local only ($\mathcal{L}_{DR}$), trained using $\mathcal{L}_{DR}$.

In Table 5, we first compare the performance of simple local models in PFL by examining $\mathcal{L}_{DR}$ and $\mathcal{L}_{CE}$. While methods using $\mathcal{L}_{CE}$ show no significant differences, utilizing $\mathcal{L}_{DR}$ leads to substantial performance improvements in PFL across all settings. The "Local only ($\mathcal{L}_{CE}$)" and "Local only ($\mathcal{L}_{CE}$ + ETF)" methods exhibit similar performance due to the nearly classwise orthogonal nature of randomly initialized classifiers (Oh et al., 2022; Saxe et al., 2013; Glorot & Bengio, 2010a; He et al., 2015; Lezama et al., 2018). With a large number of classes ($C$=100), the ETF classifier, which is also nearly classwise orthogonal, performs similarly to random initialization. When comparing **FedDr+** FT with other 2-step methods, **FedDr+** FT consistently demonstrates superior performance. This aligns with previous research (Nguyen et al., 2022; Chen et al., 2023) suggesting that fine-tuning from a well-initialized model yields better PFL performance. Additionally, compared with 1-step algorithms, **FedDr+** FT continues to show superiority, outperforming all baseline methods across all settings.

## 5 Related Work

**Federated learning.** Federated Learning (FL) is a decentralized approach to deep learning where multiple clients collaboratively train a global model using their own datasets (McMahan et al., 2017; Li et al., 2020). This approach faces challenges due to data heterogeneity across clients, causing instability in the learning process (Karimireddy et al., 2020; Luo et al., 2021). To address this problem, strategies like classifier variance reduction in FedPVR (Li et al., 2022) and virtual features in CCVR (Luo et al., 2021) have been proposed. Additionally, it is essential to distinguish between Global Federated Learning (GFL) and Personalized Federated Learning (PFL), as these are crucial concepts in FL. GFL aims to improve a single global model's performance across clients by addressing data heterogeneity through methods like client drift mitigation (Li et al., 2020; Karimireddy et al., 2020; Jhunjhunwala et al., 2023), enhanced aggregation schemes (Wang et al., 2020a;b), and data sharing techniques using public or synthesized datasets (Lin et al., 2020; Luo et al., 2021). Otherwise, PFL focuses on creating personalized models for individual clients by decoupling feature extractors and classifiers for unique updates (Oh et al., 2022; Arivazhagan et al., 2019; Collins et al., 2021), modifying local loss functions (Fallah et al., 2020; Li et al., 2021b), and using prototype communication techniques (Tan et al., 2022; Xu et al., 2023).

**Frozen classifier in FL.** By focusing on alignment, previous studies have attempted to mitigate data heterogeneity by freezing the classifier (Oh et al., 2022; Dong et al., 2022; Li et al., 2023b). Nevertheless, these methods have yet to effectively improve the alignment between features and their corresponding classifier weights. Motivated by this, we integrated the dot-regression method into FL to achieve a better-aligned local model by freezing the classifier. Dot-regression, proposed to address class imbalance, focuses on aligning feature vectors to a fixed classifier, demonstrating superior alignment performance compared to previous approaches. However, optimizing the dot-regression loss to align feature vectors with a fixed classifier caused the local model to lose information on unobserved classes, thereby degrading global model performance. To address these issues, FedLoGe (Xiao et al., 2024) employing realignment techniques to ensure the well-aligned local model's performance translated to the global model. Additionally, in FedGELA (Fan et al., 2024), the classifier is globally fixed as a simplex ETF while being locally adapted to personal distributions. Also, FedPAC Xu et al. (2023) addressed these challenges by leveraging global semantic knowledge for explicit local-global feature alignment. Besides alignment-focused methods, there have been various attempts to maintain good local model performance in the global model Jiang et al. (2023); An et al. (2024); Chen & Chao (2022).

**Knowledge distillation in FL.** Knowledge distillation (KD) has been widely studied in FL settings, such as in FedMD (Li & Wang, 2019) and FedDF (Lin et al., 2020), where a pretrained teacher model transfers knowledge to a student model. Additional distillation-based methods, such as FedFed (Yang et al., 2024) and co-distillation framework for PFL (Chen et al., 2024; Cho et al., 2023), have also been explored. In contrast to existing methods, we propose a loss function incorporating feature distillation to maintain the performance of both local and global models. To our knowledge, this is the first application of feature distillation in FL. This approach highlights the importance of distinguishing between GFL and PFL.

## 6 Limitations

While our proposed method, **FedDr+**, effectively enhances both local alignment and global knowledge preservation in Federated Learning (FL), it has certain limitations. First, our approach builds upon dot-regression to improve local alignment, but this is just one possible strategy. Alternative methods, such as directly maximizing local alignment without relying on dot-regression, could be explored to further enhance performance in FL settings. Second, although **FedDr+** effectively mitigates forgetting of unobserved classes by incorporating feature distillation, dot-regression alone remains less effective in preserving alignment for unobserved classes. While our empirical results demonstrate that **FedDr+** alleviates this issue, further theoretical investigation is needed to develop a more principled approach to ensuring alignment across both observed and unobserved classes. These limitations highlight opportunities for future work to extend and refine our method, improving its robustness and generalizability in diverse FL environments.

## 7 Conclusion

Motivated by the recent FL methods enhancing feature alignment with a fixed classifier, we first investigate the effects of applying dot-regression loss for FL. Since the dot-regression is the most direct method for feature-classifier alignment, we find it improves alignment and accuracy in local models but degrades the performance of the global model. This happens because local clients trained with dot-regression tend to forget classes that have not been observed. To address this, we propose **FedDr+**, combining dot-regression with a feature distillation method. By regularizing the deviation of local features from global features, **FedDr+** allows local models to maintain knowledge about all classes during training, thereby ultimately preserving general knowledge of the global model. Our method achieves top performance in global and personalized FL experiments, even when data is distributed unevenly across devices (non-IID settings).

## Acknowledgement

This work was supported by Institute of Information & communications Technology Planning & Evaluation (IITP) grant funded by Korea government (MSIT) [No. 2021-0-00907, Development of Adaptive and Lightweight Edge-Collaborative Analysis Technology for Enabling Proactively Immediate Response and Rapid Learning, 90%] and [No. 2019-0-00075, Artificial Intelligence Graduate School Program (KAIST), 10%].

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

---

- Appendix -

# **FedDr+**: Stabilizing Dot-regression with Global Feature Distillation for Federated Learning

---

The notations and pseudo code of **FedDr+** and **FedDr+ FT** are organized in Appendix A. In Appendix B, we provide a detailed explanation of the pulling and pushing gradients of the CE loss. In Appendix C, we provide a theoretical analysis of dot-regression, focusing on the feature vector gradient of the loss and its implications under the NTK framework, particularly for unobserved classes. The experimental setup is described in Appendix D, which includes code implementation, dataset descriptions, model specifications, optimizer settings, NIID partition, and the hyperparameter search process. Additional experimental results, including further analysis on the synergy effect and PFL, as well as results on IID dataset performance, scalability experiments, and stochastic client data settings, are presented in Appendix E.

## A   Notations, Pseudo Code of **FedDr+** and **FedDr+ FT**

In this section, we first introduce the key notations used in our method and then present the pseudocode for **FedDr+** and **FedDr+ FT**. The pseudocode provides a clear and concise implementation guide for both global federated learning (GFL) with **FedDr+** and personalized federated learning (PFL) with **FedDr+ FT**.

### A.1   Main Notations

To maintain clarity, Table 6 defines key indices, datasets, model parameters, and computations in alg and **FedDr+ FT**, forming the basis for our method and analysis.

Table 6: Notations used throughout the paper.

| **Indices** | |
|---|---|
| $c \in [C]$ | Index for a class |
| $r \in [R]$ | Index for FL round |
| $i \in [N]$ | Index for a client |
| **Dataset** | |
| $D^i_{\text{train}}$ | Training dataset for client $i$ |
| $D^i_{\text{test}}$ | Test dataset for client $i$ |
| $(x,y) \in D^i_{\text{train,test}} ; (x,y) \sim \mathcal{D}^i$ | Data on client $i$ sampled from distribution $\mathcal{D}^i$ |
| | ($x$: input data, $y$: class label) |
| $\mathcal{O}^i$ | Dataset consists of observed classes in client $i$ |
| $\mathcal{U}^i$ | Dataset consists of unobserved classes in client $i$ |
| **Parameters** | |
| $\boldsymbol{\theta}$ | Feature extractor weight parameters |
| $\boldsymbol{V} = [v_1, \ldots, v_C] \in \mathbb{R}^{C \times d}$ | Classifier weight parameters (frozen during training) |
| $v_c, c \in [C]$ | $c$-th row vector of $\boldsymbol{V}$ |
| $\Theta = (\boldsymbol{\theta}, \boldsymbol{V})$ | All model parameters |
| $\Theta^g_r = (\boldsymbol{\theta}^g_r, \boldsymbol{V})$ | Aggregated global model parameters at round $r$ |
| $\Theta^i_r = (\boldsymbol{\theta}^i_r, \boldsymbol{V})$ | Trained model parameters on client $i$ at round $r$ |
| **Model Forward** | |
| $p(x; \boldsymbol{\theta}) \in \mathbb{R}^C$ | Softmax probability of input $x$ |
| $p_c(x; \boldsymbol{\theta}), c \in [C]$ | $c$-th element of $p(x; \boldsymbol{\theta})$ |
| $\mathcal{L}_{\text{CE}}(x; \theta) = -\log p_y(x; \boldsymbol{\theta})$ | Cross-entropy loss of input $x$ |
| $f(x; \boldsymbol{\theta}) \in \mathbb{R}^d$ | Feature vector of input $x$ |
| $z(x; \boldsymbol{\theta}) = f(x; \boldsymbol{\theta})\boldsymbol{V}^\top \in \mathbb{R}^C$ | Logit vector of input $x$ |
| $z_c(x; \boldsymbol{\theta}), c \in [C]$ | $c$-th element of $z(x; \boldsymbol{\theta})$ |

### A.2  Pseudo Code of `FedDr+` and `FedDr+FT`

We now present the pseudocode for **FedDr+** and **FedDr+FT**, outlining their key operations for global and personalized federated learning. The algorithm consists of two main stages:

---

**Algorithm 1 FedDr+, FedDr+FT**

---

**Input:** Total rounds $R$, local epochs $E$, training dataset $D^i_{\text{train}}$ for client $i$, sampled client set $N^{(r)} \subset [N]$ at round $r$, learning rate $\eta^{(r)}$ at round $r$

1 **Initial Parameters:** ETF Classifier $\mathbf{V}$, Initial global model parameters $\boldsymbol{\Theta}^g_0 = (\boldsymbol{\theta}^g_0, \mathbf{V})$

2 **for** $i = 1, \dots, N$ **do**
3    Server broadcasts $\mathbf{V}$ to client $i$

4    **/** **STEP 1: Get a GFL Model $\Theta^g_R$ of FedDr+** **/**
     **for** $r = 1, \dots, R$ **do**
5      Server samples clients $N^{(r)}$ and broadcasts $\boldsymbol{\theta}^i_r \leftarrow \boldsymbol{\theta}^g_{r-1}$ **for** *each client* $i \in N^{(r)}$ ***in parallel*** **do**
6        **for** *Local Steps* $e = 1, \dots, E$ **do**
7          **for** *Batches* $j = 1, \dots, B$ **do**
8            $\boldsymbol{\theta}^i_r \leftarrow \boldsymbol{\theta}^i_r - \eta^{(r)} \nabla \mathcal{L}_{\text{Dr+}}([D^i_{\text{train}}]_j; \boldsymbol{\theta}^i_r, \boldsymbol{\theta}^g_{r-1}, \boldsymbol{V})$          *Using [Equation (1)]*
9        Upload $\boldsymbol{\theta}^i_r$ to server
10      **Server Aggregation:** $\boldsymbol{\theta}^g_r \leftarrow \frac{1}{|N^{(r)}|} \sum_{i \in N^{(r)}} \boldsymbol{\theta}^i_r$
11 **GFL output:** $\boldsymbol{\Theta}^g_R = (\boldsymbol{\theta}^g_r, \boldsymbol{V})$

12    **/** **STEP 2: Get a PFL Models $\{\Theta^i_{R+1}\}^N_{i=1}$ of FedDr+FT** **/**
     **for** $i = 1, \dots, N$ **do**
13      Server broadcasts $\boldsymbol{\theta}^i_{R+1} \leftarrow \boldsymbol{\theta}^g_R$ to client $i$
       **for** *Local Steps* $e = 1, \dots, E$ **do**
14        **for** *Batches* $j = 1, \dots, B$ **do**
15          $\boldsymbol{\theta}^i_{R+1} \leftarrow \boldsymbol{\theta}^i_{R+1} - \eta^{(R)} \nabla \mathcal{L}_{\text{Dr+}}([D^i_{\text{train}}]_j; \boldsymbol{\theta}^i_R, \boldsymbol{\theta}^g_R, \boldsymbol{V})$      *Using [Equation (1)]*

16 **PFL outputs:** $\{\boldsymbol{\Theta}^i_{R+1} = (\boldsymbol{\theta}^i_{R+1}, \boldsymbol{V})\}^N_{i=1}$

---

## B  Preliminaries: Pulling and Pushing Feature Gradients in CE

In this section, we first compute the classifier's gradient with respect to the features. Next, we explain how the cross-entropy loss draws the pulling and pushing effects.

### B.1  Feature Gradient of $\mathcal{L}_{\text{CE}}$

We begin by presenting two lemmas that support Proposition 1 and clarify pulling and pushing feature gradients in the cross-entropy (CE) loss.

**Lemma 1.** *For all* $c, c' \in [C]$, $\dfrac{\partial p_{c'}(x; \boldsymbol{\theta})}{\partial z_c(x; \boldsymbol{\theta})} = \begin{cases} p_c(x; \boldsymbol{\theta}) \cdot (1 - p_c(x; \boldsymbol{\theta})) & if\, c = c' \\ -p_c(x; \boldsymbol{\theta}) \cdot p_{c'}(x; \boldsymbol{\theta}) & otherwise \end{cases}$.

*Proof.* Note that $p(x; \boldsymbol{\theta}) = \left[ \dfrac{\exp(z_j(x; \boldsymbol{\theta}))}{\sum^C_{i=1} \exp(z_i(x; \boldsymbol{\theta}))} \right]^C_{j=1} \in \mathbb{R}^C$. Then,

(i) $c = c'$ case:

$$\frac{\partial p_c(x; \boldsymbol{\theta})}{\partial z_c(x; \boldsymbol{\theta})} = \frac{\partial}{\partial z_c(x; \boldsymbol{\theta})} \left\{ \frac{\exp(z_c(x; \boldsymbol{\theta}))}{\sum^C_{i=1} \exp(z_i(x; \boldsymbol{\theta}))} \right\} = \frac{\exp(z_c(x; \boldsymbol{\theta})) \left( \sum^C_{i=1} \exp(z_i(x; \boldsymbol{\theta})) \right) - \exp(z_c(x; \boldsymbol{\theta}))^2}{\left( \sum^C_{i=1} \exp(z_i(x; \boldsymbol{\theta})) \right)^2}$$

$$= p_c(x; \boldsymbol{\theta}) - p_c(x; \boldsymbol{\theta})^2 = p_c(x; \boldsymbol{\theta})(1 - p_c(x; \boldsymbol{\theta})).$$

(ii) $c \neq c'$ case:

$$\frac{\partial p_{c'}(x;\boldsymbol{\theta})}{\partial z_c(x;\boldsymbol{\theta})} = \frac{\partial}{\partial z_c(x;\boldsymbol{\theta})} \left\{ \frac{\exp(z_{c'}(x;\boldsymbol{\theta}))}{\sum_{i=1}^{C} \exp(z_i(x;\boldsymbol{\theta}))} \right\} = \frac{-\exp(z_c(x;\boldsymbol{\theta}))\exp(z_{c'}(x;\boldsymbol{\theta}))}{\left( \sum_{i=1}^{C} \exp(z_i(x;\boldsymbol{\theta})) \right)^2}$$

$$= -p_c(x;\boldsymbol{\theta})p_{c'}(x;\boldsymbol{\theta}).$$

$\square$

**Lemma 2.** $\nabla_{z(x;\theta)}\mathcal{L}_{CE}(x,y;\boldsymbol{\theta}) = p(x;\boldsymbol{\theta}) - \mathbf{e}_y$, *where* $\mathbf{e}_y \in \mathbb{R}^C$ *is the unit vector with its y-th element as 1.*

*Proof.*

$$\frac{\partial \mathcal{L}_{\mathrm{CE}}(x,y;\boldsymbol{\theta})}{\partial z_c(x;\boldsymbol{\theta})} = -\frac{\partial}{\partial z_c(x;\boldsymbol{\theta})} \log p_y(x;\boldsymbol{\theta}) = -\frac{1}{p_y(x;\boldsymbol{\theta})} \frac{\partial p_y(x;\boldsymbol{\theta})}{\partial z_c(x;\boldsymbol{\theta})}$$

$$= \begin{cases} p_c(x;\boldsymbol{\theta}) - 1 & \text{if } c = y \\ p_c(x;\boldsymbol{\theta}) & \text{else} \end{cases} = p_c(x;\boldsymbol{\theta}) - \mathbb{1}\{c = y\}.$$

The last equality holds by Lemma 1. Therefore, the desired result is satisfied. $\square$

**Proposition 1.** *Given* $(x,y)$, *the gradient of the* $\mathcal{L}_{CE}$ *with respect to* $f(x;\boldsymbol{\theta})$ *is given by:*

$$\nabla_{f(x;\boldsymbol{\theta})}\mathcal{L}_{CE}(x,y;\boldsymbol{\theta}) = -(1 - p_y(x;\boldsymbol{\theta}))v_y + \sum_{c \in [C]\setminus\{y\}} p_c(x;\boldsymbol{\theta})v_c \qquad (2)$$

*Proof.*

$$\nabla_{f(x;\boldsymbol{\theta})}\mathcal{L}_{\mathrm{CE}}(x,y;\boldsymbol{\theta}) = \left[ \nabla_{f(x;\boldsymbol{\theta})}z_1(x;\boldsymbol{\theta}) \mid \cdots \mid \nabla_{f(x;\boldsymbol{\theta})}z_C(x;\boldsymbol{\theta}) \right] \nabla_{z(x;\boldsymbol{\theta})}\mathcal{L}_{\mathrm{CE}}(x,y;\boldsymbol{\theta})$$

$$= \sum_{c=1}^{C} \frac{\partial \mathcal{L}_{\mathrm{CE}}(x,y;\boldsymbol{\theta})}{\partial z_c(x;\boldsymbol{\theta})} \nabla_{f(x;\boldsymbol{\theta})}z_c(x;\boldsymbol{\theta})$$

$$= \frac{\partial \mathcal{L}_{\mathrm{CE}}(x,y;\boldsymbol{\theta})}{\partial z_y(x;\boldsymbol{\theta})} \nabla_{f(x;\boldsymbol{\theta})}z_y(x;\boldsymbol{\theta}) + \sum_{c \in [C]\setminus\{y\}} \frac{\partial \mathcal{L}_{\mathrm{CE}}(x,y;\boldsymbol{\theta})}{\partial z_c(x;\boldsymbol{\theta})} \nabla_{f(x;\boldsymbol{\theta})}z_c(x;\boldsymbol{\theta})$$

$$= \frac{\partial \mathcal{L}_{\mathrm{CE}}(x,y;\boldsymbol{\theta})}{\partial z_y(x;\boldsymbol{\theta})} v_y + \sum_{c \in [C]\setminus\{y\}} \frac{\partial \mathcal{L}_{\mathrm{CE}}(x,y;\boldsymbol{\theta})}{\partial z_c(x;\boldsymbol{\theta})} v_c$$

$$= -(1 - p_y(x;\boldsymbol{\theta}))v_y + \sum_{c \in [C]\setminus\{y\}} p_c(x;\boldsymbol{\theta})v_c.$$

Applying the chain rule for the second step and invoking Lemma 2 for the final equality confirms the result.

## B.2 Physical Meaning of $\nabla_{f(x;\theta)}\mathcal{L}_{\mathbf{CE}}(x,y;\theta)$

The gradient $\nabla_{f(x;\theta)}\mathcal{L}_{\mathrm{CE}}(x,y;\boldsymbol{\theta})$ consists of two components:

$$\mathbf{F}_{\mathrm{Pull}} = \left(1 - p_y(x;\boldsymbol{\theta})\right)v_y,$$

$$\mathbf{F}_{\mathrm{Push}} = -\sum_{c \in [C]\setminus\{y\}} p_c(x;\boldsymbol{\theta})v_c.$$

$\mathbf{F}_{\mathrm{Pull}}$ moves the feature vector towards the classifier vector $v_y$ of the true class, promoting alignment. In contrast, $\mathbf{F}_{\mathrm{Push}}$ moves it away from the classifier vectors $v_c$ for $c \in [C] \setminus \{y\}$, inducing misalignment.

$\square$

# C  Theoretical Perspective of Dot-Regression (DR)

In this section, we provide a theoretical analysis of dot-regression (DR) loss in the context of feature-classifier alignment. We first derive the feature gradient of $\mathcal{L}_{\text{DR}}$ and analyze its effect on feature updates. We then present an NTK-based perspective explaining why dot-regression struggles with unobserved classes in FL. Finally, we compare DR with cross-entropy (CE) loss to highlight its limitations and the necessity of feature distillation.

## C.1  Feature Gradient of $\mathcal{L}_{\text{DR}}$

In this subsection, we derive the gradient of dot-regression loss with respect to the feature vector on the observed classes.

**Theorem C.1.** *Given $(x, y)$, the gradient of the $\mathcal{L}_{\text{DR}}$ with respect to $f(x; \boldsymbol{\theta}_f)$ is given by:*

$$\nabla_{f(x;\boldsymbol{\theta}_f)} \mathcal{L}_{\text{DR}}(x, y; \theta) = -\frac{1 - \cos\alpha}{\|f(x;\boldsymbol{\theta}_f)\|_2} \left\{ V_y - \cos\alpha \frac{f(x;\boldsymbol{\theta}_f)}{\|f(x;\boldsymbol{\theta}_f)\|_2} \right\},$$

*where $\cos\alpha = \dfrac{f(x;\boldsymbol{\theta}_f)^\top}{\|f(x;\boldsymbol{\theta}_f)\|_2} V_y$.*

*Proof.*

$$\begin{aligned}
\nabla_{f(x;\boldsymbol{\theta}_f)} \mathcal{L}_{\text{DR}}(x, y; \boldsymbol{\theta}) &= \nabla_{f(x;\boldsymbol{\theta}_f)} \left\{ \frac{1}{2} \left( \frac{f(x;\boldsymbol{\theta}_f)^T}{\|f(x;\boldsymbol{\theta}_f)\|_2} V_y - 1 \right)^2 \right\} \\
&= \left( \frac{f(x;\boldsymbol{\theta}_f)^T}{\|f(x;\boldsymbol{\theta}_f)\|_2} V_y - 1 \right) \nabla_{f(x;\boldsymbol{\theta}_f)} \frac{f(x;\boldsymbol{\theta}_f)^T}{\|f(x;\boldsymbol{\theta}_f)\|_2} V_y \\
&= \left( \frac{f(x;\boldsymbol{\theta}_f)^T}{\|f(x;\boldsymbol{\theta}_f)\|_2} V_y - 1 \right) \left[ \frac{1}{\|f(x;\boldsymbol{\theta}_f)\|_2} \left\{ I - \frac{f(x;\boldsymbol{\theta}_f) f(x;\boldsymbol{\theta}_f)^T}{\|f(x;\boldsymbol{\theta}_f)\|_2^2} \right\} V_y \right] \\
&= -\frac{1 - \cos\alpha}{\|f(x;\boldsymbol{\theta}_f)\|_2} \left\{ V_y - \cos\alpha \frac{f(x;\boldsymbol{\theta}_f)}{\|f(x;\boldsymbol{\theta}_f)\|_2} \right\}.
\end{aligned}$$

$\square$

### C.1.1  Physical Meaning of $\nabla_{f(x;\theta)} \mathcal{L}_{\text{DR}}(x, y; \theta)$

According to Theorem C.1, the change in the feature vector $\Delta f(x; \boldsymbol{\theta}_f)$ is given by:

$$\Delta f(x; \theta_f) = \eta \frac{1 - \cos\alpha}{\|f(x;\theta_f)\|_2} \left( V_y - \cos\alpha \frac{f(x;\theta_f)}{\|f(x;\theta_f)\|_2} \right),$$

where $\eta$ is the learning rate and $\alpha$ is the angle between the feature vector $f(x; \boldsymbol{\theta}_f)$ and the target vector $V_y$.

The term inside the parentheses, $V_y - \cos\alpha \frac{f(x;\theta_f)}{\|f(x;\theta_f)\|_2}$, represents a component orthogonal to $f(x; \boldsymbol{\theta}_f)$ that points towards $V_y$. This component adjusts $f(x; \boldsymbol{\theta}_f)$ to increase its cosine similarity with $V_y$ while also expanding its norm.

The scaling factor $\frac{1 - \cos\alpha}{\|f(x;\theta_f)\|_2}$ determines the update magnitude. As training progresses, $f(x; \boldsymbol{\theta}_f)$ aligns more closely with $V_y$, reducing $1 - \cos\alpha$ and increasing $\|f(x; \boldsymbol{\theta}_f)\|_2$. Consequently, $\Delta f(x; \boldsymbol{\theta}_f)$ diminishes over time, reflecting convergence as the cosine similarity with $V_y$ approaches its maximum.

Figure 8 illustrates this process, showing how the orthogonal component drives both the rotation and scaling of $f(x; \boldsymbol{\theta}_f)$ toward alignment with $V_y$.

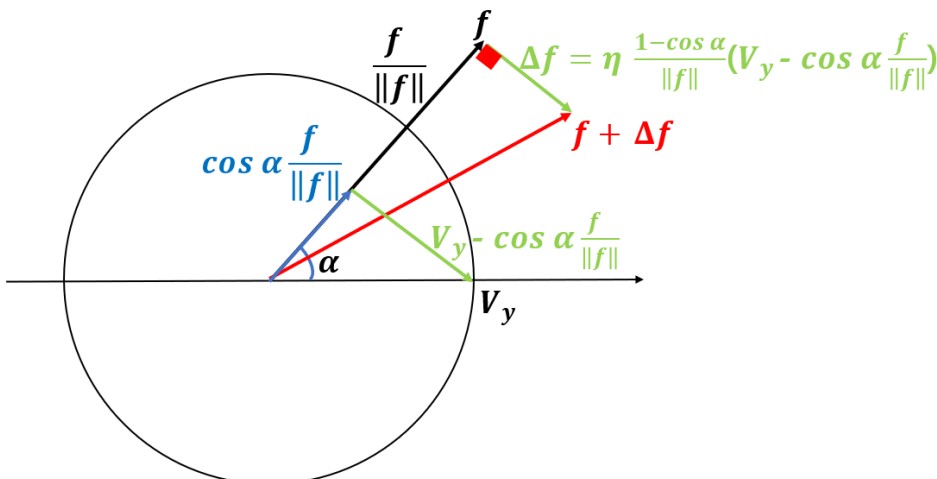

Figure 8: Feature gradient of $\mathcal{L}_{\mathrm{DR}}$. The gradient update rotates $f(x; \boldsymbol{\theta}_f)$ toward $V_y$ while increasing its norm. As training progresses, the update magnitude decreases, leading to convergence.

### C.2 NTK Perspective: Why Dot-Regression in FL Fails on Unobserved Classes

In this subsection, we analyze why dot-regression struggles with unobserved classes under the Neural Tangent Kernel (NTK) regime (Jacot et al., 2018). In the NTK regime, the feature gradient of any input is a weighted sum of the feature gradients from training samples. Assuming the network width is sufficiently wide, these weights depend only on the pair of inputs, the initialization distribution, such as He initialization (Glorot & Bengio, 2010b), and the activation functions.[§] The NTK regime holds when the setting where every layer in the neural network has infinite width, with parameters initialized i.i.d. This section explains how NTK-based gradient updates fail to align feature vectors with unobserved class directions, which leads to poor generalization in FL.

#### C.2.1 Gradient Flow in the NTK Regime

We treat gradient descent as a continuous process. $P$ is the number of trainable parameters in the feature extractor, and $\theta_p$ ($p \in [P]$) denote each parameter. We focus on a specific client, denoted by $i$.

During training, gradient descent updates the model parameters to minimize the loss function. As below, we can see the evolution of the function $f(x; \boldsymbol{\theta}(t))$ can be analyzed using the kernel $\Theta^{(L)}(t)(x, x_i)$, which evolves along the training process:

$$\frac{\mathrm{d}f(x; \boldsymbol{\theta}(t))}{\mathrm{d}t} = \sum_{p=1}^{P} \Big(\frac{\partial f(x; \boldsymbol{\theta}(t))}{\partial \theta_p}\Big)^{\top} \frac{\mathrm{d}\theta_p}{\mathrm{d}t} \qquad \text{(Chain Rule)}$$

$$= -\sum_{p=1}^{P} \Big(\frac{\partial f(x; \boldsymbol{\theta}(t))}{\partial \theta_p}\Big)^{\top} \frac{1}{|D_{\mathrm{train}}^{i}|} \sum_{(\tilde{x}, \tilde{y}) \in D_{\mathrm{train}}^{i}} \frac{\partial f(\tilde{x}; \boldsymbol{\theta}(t))}{\partial \theta_p} \nabla_{f(x_i; \theta)} \mathcal{L}(\tilde{x}, \tilde{y}; \boldsymbol{\theta}) \qquad \text{(Gradient Descent)}$$

$$= -\frac{1}{|D_{\mathrm{train}}^{i}|} \sum_{(\tilde{x}, \tilde{y}) \in D_{\mathrm{train}}^{i}} \Big(\underbrace{\sum_{p=1}^{P} \Big(\frac{\partial f(x; \boldsymbol{\theta}(t))}{\partial \theta_p}\Big)^{\top} \frac{\partial f(\tilde{x}; \boldsymbol{\theta}(t))}{\partial \theta_p}}_{\Theta^{(L)}(t)(x, \tilde{x}) \in \mathbb{R}^{d \times d}}\Big) \nabla_{f(\tilde{x}; \theta)} \mathcal{L}(\tilde{x}, \tilde{y}; \boldsymbol{\theta})$$

$$= -\frac{1}{|D_{\mathrm{train}}^{i}|} \sum_{(\tilde{x}, \tilde{y}) \in D_{\mathrm{train}}^{i}} \Theta^{(L)}(t)(x, \tilde{x}) \nabla_{f(\tilde{x}; \theta)} \mathcal{L}(\tilde{x}, \tilde{y}; \boldsymbol{\theta}) .$$

---

[§]In practice, finite-width effects cause deviations from the ideal NTK behavior.

In the NTK regime with infinitely large widths, the matrix $\boldsymbol{\Theta}^{(L)}(t)(x, \tilde{x})$ converges to a scalar multiple of the identity matrix, $\Theta_\infty^{(L)}(x, \tilde{x})\mathbf{I}$. Furthermore, with the same condition, this scalar kernel remains constant throughout training (Jacot et al., 2018; Yang, 2020; Belfer et al., 2021), though finite-width effects may introduce small variations. In the NTK regime, the gradient descent dynamics are given by:

$$\frac{\mathrm{d}f(x; \boldsymbol{\theta}(t))}{\mathrm{d}t} = -\frac{1}{|D_{\text{train}}^i|} \sum_{(\tilde{x}, \tilde{y}) \in D_{\text{train}}^i} \underbrace{\Theta_\infty^{(L)}(x, \tilde{x})}_{\in \mathbb{R}} \nabla_{f(x_i; \theta)} \mathcal{L}(\tilde{x}, \tilde{y}; \boldsymbol{\theta}) . \qquad \text{(in NTK Regime)}$$

Thus, $\Theta_\infty^{(L)}(x, \tilde{x})$ determines how each training sample $\tilde{x}$ influences an arbitrary input $x$, and in NTK regime, this weight depends only on the initialization distribution.

In Federated Learning (FL), local models are independently updated on different clients before aggregation. Under the NTK regime, each client follows the gradient flow during local training. FL aggregation then combines feature representations learned from different data distributions, leading to shifts in the global feature representation. By aggregating updates from multiple clients, FL integrates feature information from clients that have observed missing classes, thereby improving feature alignment.

### C.2.2 Limitations of Dot-Regression loss in FL under the NTK Regime

Dot-regression loss (Yang et al., 2022) speeds up the alignment of feature vectors $f(x; \boldsymbol{\theta}) \in \mathbb{R}^d$ (pre-classifier layer outputs) with the true class direction $v_y$ by minimizing the cosine angle:

$$\mathcal{L}_{\text{DR}}(x, y; \boldsymbol{\theta}, \boldsymbol{V}) = \frac{1}{2}\Big( \cos\big(f(x; \boldsymbol{\theta}), v_y\big) - 1 \Big)^2 .$$

This loss function is motivated by the decomposition of cross-entropy (CE) loss gradients into *pulling* and *pushing* components. Prior work suggests that removing the pushing effect in CE can improve convergence (Yang et al., 2022; Li & Zhan, 2021).

Let $c$ be an unobserved class for a specific client $i$ with the classifier vector $v_c$. From Theorem C.1, it follows that under the NTK regime, the gradient descent process on the client $i$ is independent of $v_c$ for arbitrary input $x$.

To analyze this, we first express the feature gradient under the dot-regression loss $\mathcal{L}_{\text{DR}}$ in the local learning stage. For simplicity, we omit the dependence on $\boldsymbol{\theta}(t)$ in the feature notation and write $\cos(f(\tilde{x}; \boldsymbol{\theta}(t)), v_y)$ as $\cos(f(\tilde{x}), v_y)$. The feature gradient is given by:

$$\frac{\mathrm{d}f(x)}{\mathrm{d}t} = \frac{1}{|D_{\text{train}}^i|} \sum_{y \in \mathcal{O}^i} \sum_{(\tilde{x}, y) \in D_{\text{train}}^i} \Theta_\infty^{(L)}(x, \tilde{x}) \frac{1 - \cos(f(\tilde{x}), v_y)}{\|f(\tilde{x})\|_2} \Big( v_y - \cos(f(\tilde{x}), v_y) \frac{f(\tilde{x})}{\|f(\tilde{x})\|_2} \Big). \quad \text{(in NTK Regime)}$$

Since $c \notin \mathcal{O}^i$, the feature gradient evaluated on training data does not depend on $v_c$. Given that feature gradients are a weighted sum over training data in the NTK regime, this implies that the learned feature representation for an arbitrary input remains unaffected by $v_c$ during local training.

Therefore, dot-regression cannot align features with unobserved classes in local training. To examine this effect more closely, consider two cases $f_1(x)$ and $f_2(x)$ with the same input $x$ with label $c$, whose settings and initialization at time $t = 0$ are identical except for the classifier vector $v_c$ of class $c$, fixed with $w$ and $-w$ ($\|w\| = 1$, $\forall y \in \mathcal{O}^i : w \perp v_y$). In the NTK regime under the dot-regression loss, we have:

$$\frac{\mathrm{d}}{\mathrm{d}t} \langle f(x), v_c \rangle = -\frac{1}{|D_{\text{train}}^i|} \sum_{y \in \mathcal{O}^i} \sum_{(\tilde{x}, y) \in D_{\text{train}}^i} \Theta_\infty^{(L)}(x, \tilde{x}) \frac{\cos(f(\tilde{x}), v_y)(1 - \cos(f(\tilde{x}), v_y))}{\|f(\tilde{x})\|_2^2} \langle f(\tilde{x}), v_c \rangle.$$

$$\text{(in NTK Regime)}$$

Since every term in the update equation is identical for $f_1(x)$ and $f_2(x)$, except for $\langle f(\tilde{x}), v_c \rangle$, which takes opposite values in each case, it follows that $\langle f_1(x), v_c \rangle = -\langle f_2(x), v_c \rangle$ for all time $t \geq 0$. This demonstrates

that classifier initialization strongly determines alignment in the local learning stage. Consequently, the global aggregation stage is the only way to generalize to classes that haven't been observed yet. This slows down the overall accuracy of the FL server.

### C.2.3   Cross-Entropy Loss and Feature-Classifier Alignment

In contrast, cross-entropy (CE) loss explicitly guides feature gradients toward $v_c$, weighted by the softmax probability $p_c$ and the NTK weight. This ensures that even when class $c$ is absent, local training still produces meaningful updates. After each global aggregation, the refined $p_c$ further strengthens alignment, allowing CE to maintain consistent feature-classifier alignment across all classes.

This observation aligns with our empirical findings: without feature distillation, dot-regression struggles to generalize to unobserved classes, whereas CE enables continuous feature updates, leading to improved generalization.

# D   Experimental Setup

This section details the code implementation, dataset descriptions, model specifications, optimizer settings, non-IID (NIID) partitioning, and hyperparameter search process used in our experiments.

## D.1   Code Implementation

Our implementations are conducted using the PyTorch framework. Specifically, the experiments presented in Table 3 and Table 4 are executed on a single NVIDIA RTX 3090 GPU, based on the code structure from the following repository: https://github.com/Lee-Gihun/FedNTD. The other parts of our study are carried out on a single NVIDIA A5000 GPU, utilizing the code framework from https://github.com/jhoon-oh/FedBABU.

## D.2   Datasets, Model, and Optimizer

To simulate a realistic FL scenario, we conduct extensive studies on three widely used datasets: CIFAR-10 (Krizhevsky et al., 2009), CIFAR-100 (Krizhevsky et al., 2009) and ImageNet-100 Deng et al. (2009). For each dataset, appropriate models are employed: VGG11 (Simonyan & Zisserman, 2014) for CIFAR-10, MobileNet (Howard et al., 2017) for CIFAR-100, and ResNet-18 (He et al., 2016) for ImageNet-100. A momentum optimizer is utilized for all experiments. The data preprocessing pipeline for the training phase includes `RandomResizedCrop`, `RandomHorizontalFlip`, and `Normalize` transformations for all datasets. During testing, only the `Normalize` transformation is applied for CIFAR-10 and CIFAR-100, while for ImageNet-100, `Resize`, `CenterCrop`, and `Normalize` are applied. Unless otherwise noted, the basic setting of our experiments follows the dataset statistics, FL scenario specifications, and optimizer hyperparameters summarized in Table 7.

Table 7: Summary of Dataset, Model, FL System, and Optimizer Specifications

| Datasets | $C$ | $|D_{\text{train}}|$ | $|D_{\text{test}}|$ | $N$ | $R$ | $r$ | $E$ | $B$ | $m$ | $\lambda$ |
|---|---|---|---|---|---|---|---|---|---|---|
| CIFAR-10 | 10 | 50000 | 10000 | 100 | 320 | 0.1 | 10 | 50 | 0.9 | 1e-5 |
| CIFAR-100 | 100 | 50000 | 10000 | 100 | 320 | 0.1 | 3 | 50 | 0.9 | 1e-5 |
| ImageNet-100 | 100 | 130000 | 5000 | 100 | 320 | 0.1 | 5 | 50 | 0.9 | 1e-5 |

Note: In terms of dataset information, $C$ represents the number of classes in the dataset, with $|D_{\text{train}}|$ and $|D_{\text{test}}|$ indicating the total numbers of training and test data used, respectively. For the federated learning (FL) system specifics, $R$ indicates the total number of FL rounds, $r$ is the ratio of clients selected for each round, and $E$ denotes the number of local epochs. Local model training utilizes a momentum optimizer where $B$ is the batch size, and $m$ and $\lambda$ represent the momentum and weight decay parameters, respectively. The initial learning rate $\eta$ is decayed by a factor of 0.1 at the 160th and 240th communication rounds. The initial learning rate $\eta$ and the number of local epochs $E$ were determined via extensive grid search for each algorithm, with details outlined in Appendix D.4.

## D.3   Non-IID Partition Strategies

To induce heterogeneity in each client's training and test data $(D_{\text{train}}^i, D_{\text{test}}^i)$, we distribute the entire class-balanced datasets, $D_{\text{train}}$ and $D_{\text{test}}$, among 100 clients using both sharding and Latent Dirichlet Allocation (LDA) partitioning strategies:

- **Sharding** (McMahan et al., 2017; Oh et al., 2022): We organize the $D_{\text{train}}$ and $D_{\text{test}}$ by label and divide them into non-overlapping shards of equal size. Each shard encompasses $\frac{|D_{\text{train}}|}{100 \times s}$ and $\frac{|D_{\text{test}}|}{100 \times s}$ samples of the same class, where $s$ denotes the number of shards per client. This sharding technique is used to create $D_{\text{train}}^i$ and $D_{\text{test}}^i$, which are then distributed to each client $i$, ensuring that each client has the same number of training and test samples. The data for each client is disjoint. As a result, each client has access to a

maximum of $s$ different classes. Decreasing the number of shards per user $s$ increases the level of data heterogeneity among clients.

- **Latent Dirichlet Allocation (LDA)** (Luo et al., 2021; Wang et al., 2020a): We utilize the LDA technique to create $D_{\text{train}}^i$ from $D_{\text{train}}$. This involves sampling a probability vector $p_c = (p_{c,1}, p_{c,2}, \cdots, p_{c,100}) \sim Dir(\alpha)$ and allocating a proportion $p_{c,k}$ of instances of class $c \in [C]$ to each client $k \in [100]$. Here, $Dir(\alpha)$ represents the Dirichlet distribution with the concentration parameter $\alpha$. The parameter $\alpha$ controls the strength of data heterogeneity, with smaller values leading to stronger heterogeneity among clients. For $D_{\text{test}}^i$, we randomly sample from $D_{\text{test}}$ to match the class frequency of $D_{\text{train}}^i$ and distribute it to each client $i$.

### D.4  Hyperparameter Search for $\eta$ and $E$

To optimize the initial learning rate ($\eta$) and the number of local epochs ($E$) for our algorithm, we conduct a grid search on the CIFAR-10, CIFAR-100, and ImageNet-100 datasets. The process and reasoning are outlined below.

#### D.4.1  Rationale for Varying Initial Learning Rate ($\eta$)

The algorithms used in our experiments differ in handling feature normalization within the loss function. Some algorithms apply feature normalization, while others do not. When features $f(x; \boldsymbol{\theta})$ are normalized, the resulting gradient is scaled by $\frac{1}{\|f(x;\boldsymbol{\theta})\|_2}$. This scaling effect necessitates a grid search across various learning rates to account for the differences in learning behavior.

#### D.4.2  Rationale for Varying Local Epochs $E$

In FL, choosing the appropriate number of local epochs is crucial. Too few epochs can lead to underfitting, while too many can cause client drift. Therefore, finding the optimal number of local epochs is essential by exploring a range of values.

#### D.4.3  Grid Search Process and Results

Considering the above reasons, we perform grid search for $\eta$ and $E$ on CIFAR-10, CIFAR-100, and ImageNet-100 datasets. The grid search for CIFAR-10 uses a shard size of 2, while for CIFAR-100, a shard size of 10 is used. Additionally, for ImageNet-100, a shard size of 20 is used. The detailed procedures for each dataset are provided below. These optimal settings have also been confirmed to yield good performance in less heterogeneous settings.

**CIFAR-10.** We examine $\eta$ values from $\{0.01, 0.05, 0.1, 0.15, 0.2, 0.25, 0.3, 0.35, 0.4, 0.45, 0.5, 0.55, 0.6\}$. For $E$, we consider $\{1, 3, 5, 10, 15\}$. A default initial learning rate of 0.01 is used unless specified otherwise. The optimal learning rates vary by algorithm, and the results are summarized in Table 8. Table 8 also includes the additional hyperparameters used for each algorithm. The notation for these additional hyperparameters follows the conventions used throughout this paper (Li et al., 2020; Lee et al., 2022; Jhunjhunwala et al., 2023; Li et al., 2023b; Lee et al., 2024). The optimal number of local epochs is found to be 10 for every algorithm.

**CIFAR-100.** We examine $\eta$ values from $\{0.1, 0.5, 1.0, 1.5, 2.0, 2.5, 3.0, 3.5, 4.0, 4.5, 5.0, 5.5, 6.0, 6.5, 7.0\}$. For $E$, we consider $\{1, 3, 5, 10\}$. A default initial learning rate of 0.1 is used unless specified otherwise. The optimal learning rates differ by algorithm, and the results are listed in Table 9. Table 9 also includes the additional hyperparameters used for each algorithm. The notation for these additional hyperparameters follows the conventions used throughout this paper (Li et al., 2020; Lee et al., 2022; Jhunjhunwala et al., 2023; Li et al., 2023b; Lee et al., 2024). The optimal number of local epochs is found to be 3 for every algorithm.

**ImageNet-100.** We examine $\eta$ values from $\{0.01, 0.1, 1.0, 10.0\}$, which are chosen to maintain a consistent logarithmic scale difference. A default initial learning rate of 0.1 is used unless specified otherwise. The

optimal learning rates differ by algorithm, and the results are listed in Table 10. Table 10 also includes the additional hyperparameters used for each algorithm. The notation for these additional hyperparameters follows the conventions used throughout this paper (Li et al., 2020; Lee et al., 2022; Jhunjhunwala et al., 2023; Li et al., 2023b; Lee et al., 2024). The optimal number of local epochs is fixed at 5, following the setting of (Lee et al., 2024).

Table 8: Hyperparameters for VGG11 training on CIFAR-10.

| Hyperparameters | Feature un-normalized algorithms | | | | | | | | | Feature normalized algorithms | | |
|---|---|---|---|---|---|---|---|---|---|---|---|---|
| | FedAvg | FedBABU | FedProx | SCAFFOLD | MOON | FedNTD | FedExP | FedSOL | FedGELA | FedETF | SphereFed | **FedDr+** (Ours) |
| $\eta$ | 0.01 | 0.01 | 0.01 | 0.01 | 0.01 | 0.01 | 0.01 | 0.01 | 0.01 | 0.05 | 0.55 | 0.35 |
| Additional | None | None | $\mu$=0.001 | None | $(\mu, \tau)$=(1,0.5) | $(\beta, \tau)$=(1,3) | $\epsilon$=0.001 | $\rho$=2.0 | None | $(\beta, \tau)$=(1,1) | None | $\beta$=0.9 |

Table 9: Hyperparameters for MobileNet training on CIFAR-100.

| Hyperparameters | Feature un-normalized algorithms | | | | | | | | | Feature normalized algorithms | | |
|---|---|---|---|---|---|---|---|---|---|---|---|---|
| | FedAvg | FedBABU | FedProx | SCAFFOLD | MOON | FedNTD | FedExP | FedSOL | FedGELA | FedETF | SphereFed | **FedDr+** (Ours) |
| $\eta$ | 0.1 | 0.1 | 0.1 | 0.1 | 0.1 | 0.1 | 0.1 | 0.1 | 0.1 | 0.5 | 6.5 | 5.0 |
| Additional | None | None | $\mu$=0.001 | None | $(\mu, \tau)$=(1,0.5) | $(\beta, \tau)$=(1,3) | $\epsilon$=0.001 | $\rho$=2.0 | None | $(\beta, \tau)$=(1,1) | None | $\beta$=0.9 |

Table 10: Hyperparameters for ResNet-18 training on ImageNet-100.

| Hyperparameters | Feature un-normalized algorithms | | | | | | | | | Feature normalized algorithms | | |
|---|---|---|---|---|---|---|---|---|---|---|---|---|
| | FedAvg | FedBABU | FedProx | SCAFFOLD | MOON | FedNTD | FedExP | FedSOL | FedGELA | FedETF | SphereFed | **FedDr+** (Ours) |
| $\eta$ | 0.1 | 0.1 | 0.1 | 0.1 | 0.1 | 0.1 | 0.1 | 0.1 | 0.1 | 1.0 | 1.0 | 1.0 |
| Additional | None | None | $\mu$=0.001 | None | $(\mu, \tau)$=(1,0.5) | $(\beta, \tau)$=(1,3) | $\epsilon$=0.001 | $\rho$=2.0 | None | $(\beta, \tau)$=(1,1) | None | $\beta$=0.9 |

# E   Additional Experiment Results

This section provides additional experimental results, including analysis on the synergy effect, personalized FL (PFL), IID dataset performance, stochastic client data settings, and scalability experiments.

## E.1   Synergy Effect Details

Table 11: Synergy of various FL algorithms and regularizers. Baseline indicates training FL models without a regularizer. FD denotes feature distillation, which is the regularizer we use in **FedDr+**.

| Algorithm | Sharding ($s = 10$) | | | | | | | LDA ($\alpha = 0.1$) | | | | | | |
| --- | --- | --- | --- | --- | --- | --- | --- | --- | --- | --- | --- | --- | --- | --- |
| | Baseline | +Prox | +MOON | +KD | +NTD | +LD | +FD | Baseline | +Prox | +MOON | +KD | +NTD | +LD | +FD |
| FedAvg | 37.22 | 36.87 | 37.43 | 36.25 | 37.71 | 37.17 | 37.82 | 42.52 | 43.22 | 44.79 | 44.21 | 43.39 | 43.43 | 43.76 |
| FedBABU | 46.20 | 46.03 | 46.49 | 46.37 | 47.22 | 46.71 | 46.95 | 47.37 | 46.62 | 46.27 | 47.60 | 46.48 | 45.78 | 46.49 |
| SphereFed | 43.90 | 41.96 | 43.13 | 44.94 | 43.47 | 43.95 | 45.21 | 46.98 | 43.77 | 46.81 | 47.76 | 47.25 | 47.01 | 49.74 |
| FedETF | 32.42 | 31.87 | 34.30 | 32.76 | 32.65 | 32.25 | 32.77 | 46.27 | 45.71 | 45.98 | 46.67 | 46.16 | 45.91 | 46.47 |
| FedGELA | 29.17 | 28.69 | 28.80 | 29.11 | 28.84 | 29.36 | 30.33 | 27.11 | 29.03 | 28.09 | 28.45 | 29.62 | 29.41 | 29.75 |
| Dot-Regression | 42.52 | 41.95 | 44.72 | 47.45 | 48.32 | 47.52 | **48.69** | 42.72 | 46.35 | 50.36 | 49.47 | 50.36 | 49.28 | **50.86** |

Table 12: Optimal $\beta$ value selected through grid search to achieve the best synergy of various FL algorithms and regularizers.

| Algorithm | Sharding ($s = 10$) | | | | | | | LDA ($\alpha = 0.1$) | | | | | | |
| --- | --- | --- | --- | --- | --- | --- | --- | --- | --- | --- | --- | --- | --- | --- |
| | Baseline | +Prox | +MOON | +KD | +NTD | +LD | +FD | Baseline | +Prox | +MOON | +KD | +NTD | +LD | +FD |
| FedAvg | None | 0.999 | 0.5 | 0.9999 | 0.9999 | 0.999 | 0.9 | None | 0.999 | 0.99 | 0.999 | 0.99 | 0.999 | 0.9999 |
| FedBABU | None | 0.9999 | 0.9 | 0.999 | 0.99 | 0.999 | 0.999 | None | 0.999 | 0.999 | 0.999 | 0.999 | 0.99 | 0.9999 |
| SphereFed | None | 0.9999 | 0.9999 | 0.9999 | 0.9 | 0.99 | 0.9 | None | 0.9999 | 0.999 | 0.999 | 0.9999 | 0.999 | 0.99 |
| FedETF | None | 0.999 | 0.3 | 0.5 | 0.999 | 0.9 | 0.9 | None | 0.9999 | 0.9999 | 0.5 | 0.999 | 0.99 | 0.99 |
| FedGELA | None | 0.9999 | 0.7 | 0.5 | 0.7 | 0.5 | 0.7 | None | 0.99 | 0.9 | 0.5 | 0.5 | 0.5 | 0.3 |
| Dot-Regression | None | 0.9999 | 0.9 | 0.5 | 0.5 | 0.5 | 0.9 | None | 0.9999 | 0.5 | 0.5 | 0.5 | 0.5 | 0.9 |

Table 13: Synergy of various FL algorithms and regularizers at $\beta = 0.9$.

| Algorithm | Sharding ($s = 10$) | | | | | | | LDA ($\alpha = 0.1$) | | | | | | |
| --- | --- | --- | --- | --- | --- | --- | --- | --- | --- | --- | --- | --- | --- | --- |
| | Baseline | +Prox | +MOON | +KD | +NTD | +LD | +FD | Baseline | +Prox | +MOON | +KD | +NTD | +LD | +FD |
| FedAvg | 37.22 | 30.27 | 36.67 | 35.14 | 35.56 | 34.83 | 37.82 | 42.52 | 36.09 | 42.09 | 41.48 | 41.34 | 43.36 | 43.10 |
| FedBABU | 46.20 | 36.71 | 46.49 | 45.50 | 45.09 | 45.81 | 45.31 | 47.37 | 39.04 | 45.92 | 45.58 | 45.56 | 46.46 | 44.77 |
| SphereFed | 43.90 | 1.36 | 1.89 | 41.01 | 43.47 | 41.73 | 45.21 | 46.98 | 1.46 | 2.21 | 45.22 | 46.25 | 43.84 | 48.61 |
| FedETF | 32.42 | 25.18 | 32.58 | 32.76 | 31.98 | 32.25 | 32.77 | 46.27 | 34.92 | 45.38 | 44.94 | 45.77 | 44.36 | 45.92 |
| FedGELA | 29.17 | 25.52 | 28.57 | 28.84 | 28.67 | 28.37 | 29.07 | 27.11 | 26.84 | 28.09 | 27.78 | 28.27 | 27.97 | 27.60 |
| Dot-Regression | 42.52 | 5.42 | 44.72 | 46.60 | 45.78 | 47.52 | **48.69** | 42.72 | 7.47 | 30.69 | 48.19 | 33.08 | 49.09 | **50.79** |

We evaluate the synergy effect of various FL algorithms by maintaining their original training loss while incorporating specific regularizers, as detailed in Equation 1 of the main text. To manage the differing loss scales between the baseline FL algorithms and the regularizers, we systematically tune the coefficient $\beta$ across a range of values $(0.1, 0.3, 0.5, 0.7, 0.9, 0.99, 0.999, 0.9999)$. The resulting performance and optimal $\beta$ values are shown in Table 11 and Table 12. However, when we set $\beta = 0.9$ without addressing the issue of differing loss scales, the performance results, presented in Table 13, reveal that several synergies are significantly inferior due to this oversight.

Table 14: PFL accuracy comparison with MobileNet on CIFAR-100. For PFL, we denote the entries in the form of $X_{\pm(Y)}$, representing the mean and standard deviation of personalized accuracies across all clients derived from a single seed.

| Algorithm | $s{=}10$ | $s{=}20$ | $s{=}100$ | $\alpha{=}0.05$ | $\alpha{=}0.1$ | $\alpha{=}0.3$ |
|---|---|---|---|---|---|---|
| Dot-Regression | 42.52 | 49.02 | 52.86 | $30.31_{\pm 7.95}$ | $37.52_{\pm 5.60}$ | $47.08_{\pm 3.69}$ |
| Dot-Regression FT ($\mathcal{L}_{DR}$) | $80.84_{\pm(5.99)}$ | $74.18_{\pm(5.78)}$ | $56.84_{\pm(5.04)}$ | $72.02_{\pm(6.80)}$ | $66.96_{\pm(5.36)}$ | $60.34_{\pm(3.66)}$ |
| Dot-Regression FT ($\mathcal{L}_{Dr+}$) | $80.82_{\pm(6.12)}$ | $73.73_{\pm(5.75)}$ | $56.69_{\pm(4.95)}$ | $71.85_{\pm(7.03)}$ | $66.59_{\pm(5.32)}$ | $59.87_{\pm(3.65)}$ |
| **FedDr+** (ours) | 48.69 | 51.00 | 53.23 | $39.63_{\pm 9.12}$ | $45.83_{\pm 6.18}$ | $48.04_{\pm 3.44}$ |
| **FedDr+** FT ($\mathcal{L}_{DR}$) (ours) | $\mathbf{84.23}_{\pm(5.44)}$ | $\mathbf{75.73}_{\pm(4.79)}$ | $\mathbf{56.90}_{\pm(4.85)}$ | $\mathbf{78.65}_{\pm(6.17)}$ | $\mathbf{74.86}_{\pm(4.77)}$ | $\mathbf{62.47}_{\pm(3.72)}$ |
| **FedDr+** FT ($\mathcal{L}_{Dr+}$) (ours) | $84.10_{\pm(5.43)}$ | $75.42_{\pm(4.80)}$ | $56.76_{\pm(4.91)}$ | $78.55_{\pm(6.16)}$ | $74.75_{\pm(4.75)}$ | $62.16_{\pm(3.73)}$ |

## E.2 Personalized Federated Learning Results

We introduce **FedDr+** FT and dot-regression FT, inspired by prior work (Oh et al., 2022; Dong et al., 2022; Li et al., 2023b; Kim et al., 2023). These methods enhance personalization by leveraging local data to fine-tune the GFL model. We investigate the impact of fine-tuning using $\mathcal{L}_{Dr+}$ and $\mathcal{L}_{DR}$ loss for each GFL model to assess their effectiveness on personalized accuracy. Performance metrics without standard deviations indicate results on $D_{\text{test}}$, obtained from the GFL model after the initial step in the 2-step method. Our experiments involve heterogeneous settings with sharding and LDA non-IID environments, using MobileNet on CIFAR-100 datasets. We set $s$ as 10, 20, and 100, and the LDA concentration parameter ($\alpha$) as 0.05, 0.1, and 0.3. Table 14 provides detailed personalized accuracy results.

Our 2-step process involves first developing the GFL model either using dot-regression or **FedDr+**. In the second step, we fine-tune this model to create the PFL model, again using $\mathcal{L}_{DR}$ or $\mathcal{L}_{Dr+}$. This results in four combinations: Dot-Regression FT ($\mathcal{L}_{DR}$), Dot-Regression FT ($\mathcal{L}_{Dr+}$), **FedDr+** FT ($\mathcal{L}_{DR}$), and **FedDr+** FT ($\mathcal{L}_{Dr+}$). When the GFL model is fixed, using $\mathcal{L}_{DR}$ for fine-tuning consistently outperforms $\mathcal{L}_{Dr+}$ across all settings, because dot-regression focuses on local alignment which advantages personalized fine-tuning. Conversely, when the fine-tuning method is fixed, employing $\mathcal{L}_{Dr+}$ for the GFL model consistently outperforms $\mathcal{L}_{DR}$ across all settings. This aligns with previous research (Nguyen et al., 2022; Chen et al., 2023) suggesting that fine-tuning from a well-initialized model yields better PFL performance.

## E.3 IID Data Performance

To address the question regarding the performance of **FedDr+** or dot-regression loss in Federated Learning (FL) settings with IID data, we conducted experiments on CIFAR-100 with 100 clients, distributing data IID and ensuring a fair number of samples per client. We evaluated FedAvg, FedBABU, Dot-regression, and **FedDr+** across 5 seeds, calculating the mean and standard deviation of the global model accuracy for each algorithm.

Table 15: Global model accuracy (%) in IID data settings.

| Algorithm | Accuracy (mean $\pm$ std) |
|---|---|
| **FedAvg** | $47.19 \pm 1.06$ |
| **FedBABU** | $45.18 \pm 0.61$ |
| **Dot-regression** | $51.48 \pm 0.99$ |
| **FedDr+** | $\mathbf{51.10 \pm 0.61}$ |

From the Table 15, it is evident that Dot-regression and **FedDr+** achieve the highest performance, significantly outperforming both FedAvg and FedBABU. The performance of Dot-regression and **FedDr+** is nearly identical under IID settings.

This similarity arises because, in the IID scenario, there are no **unobserved classes** across clients. As a result, the feature distillation mechanism in **FedDr+**, which is specifically designed to mitigate forgetting on unobserved classes, does not provide additional benefits. Instead, both Dot-regression and **FedDr+** excel in improving local alignment across all classes, fully achieving the global model's objective of enhancing local alignment for all clients.

### E.4    Performance in Stochastic Client Data Settings

While our original experiments on **CIFAR-100 (s=10) with 100 clients** assumed a static client dataset, we conducted additional experiments where each client randomly removed one class from its dataset every 10 FL rounds. As expected, global model accuracy decreased for all methods, as shown in Table 16. However, **FedDr+** consistently outperformed CE and Dot-regression, demonstrating its robustness in handling dynamic class distributions. The round-wise global test accuracy trends for CE, Dot-regression, and **FedDr+** in the stochastic setting are presented in Figure 9c, further confirming **FedDr+**'s stability and superior performance across training rounds.

Table 16: Global model accuracy (%) in static and stochastic client data settings.

| Algorithm | Static Setting | Stochastic Setting |
|---|---|---|
| **CE** | 46.20 | 43.59 |
| **Dot-regression** | 42.52 | 38.13 |
| **FedDr+** | **48.69** | **44.96** |

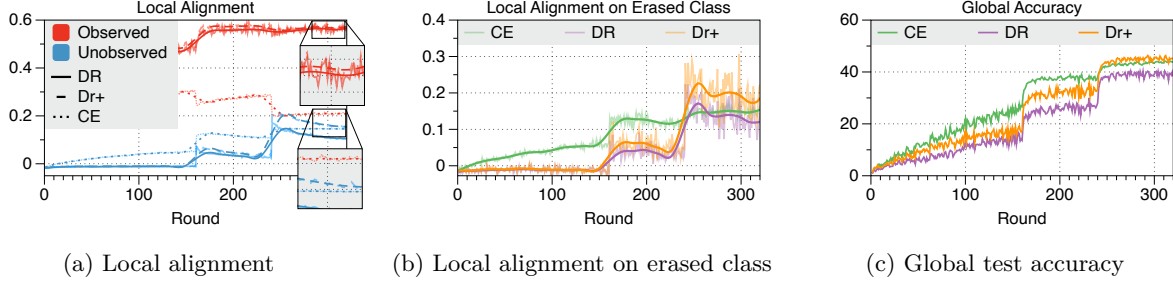

(a) Local alignment          (b) Local alignment on erased class          (c) Global test accuracy

Figure 9: Comparison of (a) feature-classifier alignment on the observed and unobserved classes test data, (b) feature-classifier alignment on erased-class test data for $\boldsymbol{\theta}_r^i$, and (c) global test accuracy of $\boldsymbol{\theta}_r^g$ on all classes. Models are trained using $\mathcal{L}_{\text{CE}}$, $\mathcal{L}_{\text{DR}}$, and $\mathcal{L}_{\text{Dr+}}$.

To further investigate why **FedDr+** maintains superior global accuracy in the stochastic setting, we analyzed the feature-classifier alignment for both observed/unobserved classes and erased classes.

- **Local alignment for observed/unobserved classes (Fig 9a):**
  - **FedDr+** maintains superior feature-classifier alignment for both observed and unobserved classes compared to Dot-regression, consistently outperforming it across all rounds.
  - During the final convergence phase, **FedDr+** surpasses even CE in unobserved class alignment, confirming its effectiveness in preserving global knowledge.

- **Local alignment for erased class (Fig 9b):**
  - Even for erased class (those removed during training), **FedDr+** retains stronger feature-classifier alignment than Dot-regression.
  - During the final convergence phase, **FedDr+** also surpasses CE in erased class alignment, further demonstrating its ability to mitigate forgetting of removed class knowledge.

These results suggest that the **feature distillation mechanism in `FedDr+` effectively enhances global knowledge preservation while also enabling effective learning of observed classes, even when class distributions change dynamically**.

### E.5 Scaling to Larger Numbers of Clients and Training Rounds

We conducted experiments on **CIFAR-100 (s=10)** with **1,000 communication rounds**, increasing the number of clients to 100, 200, 500, and 1,000. All algorithms used previously grid-searched optimal hyperparameters, and results are averaged over three independent seeds. All algorithms used previously grid-searched optimal hyperparameters, and results are averaged over three independent seeds.

Table 17: Global model accuracy (%) for different numbers of clients with 1,000 communication rounds.

| Algorithm | N=100 | N=200 | N=500 | N=1,000 |
|---|---|---|---|---|
| **FedAvg** | $50.50 \pm 0.57$ | $42.51 \pm 1.47$ | $33.02 \pm 0.74$ | $26.63 \pm 1.31$ |
| **FedBABU** | $58.19 \pm 1.07$ | $48.75 \pm 1.99$ | $37.40 \pm 0.41$ | $25.10 \pm 1.08$ |
| **FedDr+** | $\mathbf{64.21 \pm 1.24}$ | $\mathbf{59.78 \pm 0.71}$ | $\mathbf{43.27 \pm 0.31}$ | $\mathbf{28.99 \pm 0.98}$ |

Table 17 confirms that **`FedDr+`** consistently outperforms FedAvg and FedBABU across all settings, demonstrating robust scalability in large-scale FL.

