# OpenReview forum: "FedDr+: Stabilizing Dot-regression with Global Feature Distillation for Federated Learning"
_TMLR — Accepted by TMLR_

### Review · Reviewer_BYv6 · 2024-12-22

**Summary Of Contributions:**

The paper proposes a mechanism for solving the two important problems of locally optimized FL (personalized FL) and maintaining global knowledge (global FL). By leveraging the dot-regression loss function alongside other enhancements, the article generates a novel FL-compatible loss function geared towards the objective above. The dot-regression loss ensures that the feature vector per input and the corresponding classifier parameters vector have a similar alignment (in terms of their angle measured by cosine). Such a method prompts the development of classifiers in tune with the local client’s features. Furthermore, to ensure that the local models maintain higher predictivity for unobserved classes (not available in client training data) the global model features are distilled to the client features by means of a mean squared error loss function between the global and local model parameters. Overall, combining these two loss functions in a weighted fashion provides the base loss function for both the global and local FL systems. Finally, experiments on CIFAR-10/100 and ImageNet-100 demonstrate the empirical efficacy of both approaches FedDr+ and FedDr+FT for global and local FL respectively.

**Audience:**

Yes

**Claims And Evidence:**

Yes

**Requested Changes:**

The changes requested are highlighted in the weaknesses section above. Improving the overall writing, and identifying the convergence analysis for the newly proposed FL method alongside quantifying the impact of FL on privacy are the main suggested changes.

**Strengths And Weaknesses:**

Strengths:
1. The article provides interesting (and new for the FL system) loss functions that enhance feature alignment in FL (for the local models) while further ensuring global knowledge is not lost.
2. The two major proposed loss functions are simple to add to the existing objectives ensuring practical use cases.
3. Furthermore, both proposed objectives are intuitive to understand in the context of the current problem.
4. Experimental results verify the utility of the proposed solutions.

Weaknesses:
1. The paper is hard to read. Although there is sufficient focus on FL systems and related nomenclature, some key topics such as local alignment are not properly elaborated upon. Ideally, adding a summary about local alignment and global knowledge preservation is necessary to improve the paper’s readability.
2. Furthermore, it is unclear from the article whether the final aim is to solve local and global problems simultaneously or via two different models. For example, if we learn two models from the same FL setup, why are such two models not superior to a single solution? Intuition about the ‘why’ behind the proposed primary problem would be helpful to further appreciate the paper’s contributions.
3. Can the authors elaborate upon the exact changes required for demonstrating the convergence of the newly proposed federated algorithm? I believe the primary updates are to the loss function, but are there certain important changes to how the algorithm converges in comparison to FedAvg (or similar algorithms)?
4. One of the use cases of FL is to limit the data sent to the server and provide some form of data-minimization privacy. Given the form of the new loss function incorporated in the article, can the authors elaborate on the privacy impact of the updated method? Furthermore, it is the new method trivially extensible to differential privacy or other aggregation methods such as secure party computation?

---

> ### Author Response · Authors · 2025-02-04
> **Response to Reviewer BYv6 (1/2)**
>
> We sincerely appreciate your detailed feedback and constructive suggestions. Below, we provide responses to each of your concerns.
>
> ---
>
> ### Q1) Clarification on Local Alignment and Global Knowledge Preservation
>
> **A1)** We have updated the **Introduction** to clearly define local alignment and global knowledge preservation. **Local alignment** is now described as cosine similarity between the features extracted by the local model and the classifier’s true class vectors, computed on the client’s training data, aiming to maximize alignment for improved local training. **Global knowledge preservation** is defined as retaining the global model’s knowledge of rare or unobserved classes in the client’s training data, preventing forgetting during local training.
>
> We hope this update addresses your concern. Please let us know if further clarification is needed.
>
>
> ### Q2) Clarification on GFL and PFL Relationship
>
> **A2)** The distinction between global and personalized federated learning (GFL and PFL) is fundamental. Below, we clarify why these problems are inherently distinct and how our proposed method addresses them effectively.
>
> 1. **GFL and PFL Are Fundamentally Different Problems**
>    - **GFL** aims to create a well-generalized global model that performs effectively across all clients by balancing knowledge from diverse, non-iid data distributions.
>    - **PFL**, on the other hand, focuses on tailoring models to individual clients’ unique data, optimizing for client-specific data distributions.
>    These objectives are inherently conflicting; a single model cannot simultaneously achieve both global generalization and client-specific personalization due to the trade-off between the two goals.
>
> 2. **How FedDr+ and FedDr+FT Address These Problems**
>    - **FedDr+** is designed for **GFL** by addressing both **local alignment** and **global knowledge preservation**:
>      - **Dot-regression loss** improves **local alignment**, ensuring that local models effectively align features with classifier vectors for stable training.
>      - **Feature distillation** preserves **global knowledge**, preventing the global model from forgetting **rare and unobserved classes**.
>
>      By combining these mechanisms, **FedDr+ produces a well-generalized global model**, balancing **local adaptation** and **global retention**. The output of FedDr+ is a **single global model** shared across clients.
>    - **FedDr+FT** extends FedDr+ to **PFL** by fine-tuning the FedDr+ global model for client-specific data using the same FedDr+ loss. This approach leverages the strong initialization from FedDr+ to achieve effective personalization while preserving the benefits of the global model. The output of FedDr+FT is a set of personalized models.
>
> 3. **Why a Two-Stage Framework Is Effective**
>    - FedDr+ and FedDr+FT address GFL and PFL sequentially, acknowledging their distinct objectives. By starting with a well-trained global model from FedDr+, FedDr+FT ensures effective personalization without compromising the global model’s generalization capabilities. This aligns with previous research[1,2], which highlights that fine-tuning from a well-initialized global model significantly improves PFL performance. This two-stage approach demonstrates the necessity of separating GFL and PFL objectives while leveraging the synergy between them through strong model initialization.
>
> 4. **Revisions to Enhance Clarity**
>    - Clarified in the **Introduction** how FedDr+FT extends FedDr+ to PFL, emphasizing the importance of starting with a well-generalized global model for effective personalization. .
>    - Added pseudocode for FedDr+ and FedDr+FT in **Appendix A** for a clear implementation guide.
>
>    We hope this update addresses your concern. Please let us know if further clarification is needed.

---

> ### Author Response · Authors · 2025-02-04
> **Response to Reviewer BYv6 (2/2)**
>
> ### Q3) Response to Convergence-Related Questions
>
> **A3)** To address the question on convergence, we analyze the key differences in how FedDr+ achieves convergence compared to FedAvg and similar algorithms.
>
> 1. Advancements Over FedAvg and FedAvg+ETF Classifier Freezing
>    - Both **FedAvg** and **FedAvg+ETF Classifier Freezing** use CE loss.
>    - The key difference is that **FedAvg+ETF Classifier Freezing** stabilizes local alignment by ensuring features extracted from client data align with a shared frozen ETF classifier. This setup improves performance in high-heterogeneity settings (s=10) compared to standard FedAvg, which does not freeze the classifier.
>
> 2. Dot-Regression Loss and Its Limitations
>    - **Dot-regression loss**, compared to CE loss, optimizes local alignment more effectively, achieving faster convergence for observed classes.
>    - However, its lack of robustness to unobserved classes results in degraded global model performance.
>
> 3. FedDr+ (Dot-Regression with Feature Distillation)
>    - **FedDr+** overcomes the limitations of dot-regression loss by incorporating feature distillation, which preserves global knowledge and improves alignment for unobserved classes.
>    - This enables FedDr+ to achieve superior global model performance compared to CE-based methods and standalone dot-regression.
>
> Our experiments show that **FedDr+** outperforms both **FedAvg** and **FedAvg+ETF Classifier Freezing**, achieving faster convergence for observed classes and improved local alignment for unobserved classes. This leads to a well-generalized global model with superior accuracy, especially in high-heterogeneity settings.
>
>
> ### Q4) Response to Privacy Impact and Extensibility
> **A4)** Below is our explanation that aligns with the privacy impact and extensibility:
> #### **Privacy Impact of the Proposed Method**
>
> Our method follows standard Federated Learning (FL) principles, ensuring that **raw data remains on clients** and only model updates are shared with the server. Notably, **freezing the classifier** removes the need to transmit classifier parameters across rounds, which:
>
> - **Reduces communication overhead**, as classifier updates typically contribute significantly to transmitted model parameters.
> - **Mitigates potential privacy risks** related to class imbalance, where **weight norm biases** in classifier updates might indirectly reveal client-specific class distributions.
>
> By eliminating classifier updates, FedDr+ inherently reduces the amount of shared model updates, contributing to a more communication-efficient and privacy-preserving FL system.
>
> #### **Compatibility with Differential Privacy (DP) and Secure Aggregation**
>
> FedDr+ is fully compatible with Differential Privacy (DP) and Secure Aggregation without requiring algorithmic modifications. Specifically:
>
> - **Differential Privacy (DP):** DP mechanisms, such as **local DP** (adding noise before transmission) or **server-side DP aggregation**, can be applied directly to FedDr+ as in standard FL. Since our method does not modify the update structure, **DP noise addition remains straightforward**.
> - **Secure Aggregation:** FedDr+ is directly compatible with encryption-based secure aggregation protocols. By **not transmitting classifier updates**, our method **reduces the size of transmitted updates**, making **encryption and computational costs more efficient**.
>
> Thus, FedDr+ seamlessly integrates with existing privacy-preserving FL techniques, maintaining strong privacy guarantees while improving efficiency and stability in non-IID settings.
>
>
> ---
> ### **Manuscript Revisions**
>
> We have revised the manuscript as follows:
>
> - **Introduction**:
>   - Clarified the definition of **local alignment** and **global knowledge preservation** for improved understanding.
>   - Provided a structured explanation of **how FedDr+FT extends FedDr+ to PFL**, emphasizing the role of a well-generalized global model for effective personalization.
>
> - **Appendix A**:
>   - Added **pseudocode for FedDr+ and FedDr+FT** to provide a clear implementation guide.
>
> These revisions provide improved clarity.
>
> ---
> ### **References**
>
> [1] Where to begin? on the impact of pre-training and initialization in federated learning, NeurIPS 2022
>
> [2] On the importance and applicability of pre-training for federated learning, ICLR 2023

---

### Review · Reviewer_TUZs · 2024-12-24

**Summary Of Contributions:**

The paper proposed FedDr+ which combines the dot-regression loss [1] with Feature Distillation [2] for federated learning over non-iid data. The main idea of the algorithm is as follows:
1. Initialize the classifier to satisfy the simplex Equiangular Tight Frame (ETF) condition and freeze it.
2. Train the remaining network using FedAvg algorithm where the local loss on each client is given as $\beta L_{DR} + (1-\beta) L_{FD}$.

    a. $L_{DR}$ is the dot regression loss which tries to minimize the cosine difference between the feature extractor output $f(x,\theta)$ and the classifier's weight vector corresponding to the ground truth class $v_y$. Here, $\theta$ is the trainable parameters set and $V$ is frozen i.e., only the feature extractor is trained during back-propagation.

    b. $L_{FD}$ is the feature distillation loss which reduces the $L_2$ distance between the  global model’s feature vectors $f(x, \theta^g_{r-1})$ and local model's feature vector  $f(x, \theta^i_{r})$ at each client $i$. This regularizes the local feature vectors to be closer to the received global feature vectors at each round.

The experimental results show that combining dot regression loss with feature distillation loss results in performance improvements for FL with non-IID data in both global and personalized settings.

References:
1. Yibo Yang, Shixiang Chen, Xiangtai Li, Liang Xie, Zhouchen Lin, and Dacheng Tao. Inducing neural collapse in imbalanced learning: Do we really need a learnable classifier at the end of deep neural network? Advances in Neural Information Processing Systems, 35:37991–38002, 2022
2. Byeongho Heo, Jeesoo Kim, Sangdoo Yun, Hyojin Park, Nojun Kwak, and Jin Young Choi. A comprehensive overhaul of feature distillation. In Proceedings of the IEEE/CVF International Conference on Computer Vision, pp. 1921–1930, 2019.

**Audience:**

Yes

**Claims And Evidence:**

Yes

**Requested Changes:**

1. Including the results on IID data setting can provide valuable insights into the proposed FedDr+
2. Please add a limitation section addressing the trade-offs of the proposed algorithm.
3. Conducting experiments on multiple seeds and reporting the mean and standard deviation of the accuracy is important as FL with non-IID data usually has high variance in the performance.

Minor changes:
1. Add a pseudo code for FedDr+ in the appendix to provide a concise implementation guide for the readers.

**Strengths And Weaknesses:**

Strengths:
1. The paper investigates the effects of applying dot-regression loss for FL
2. The paper proposes FedDr+ algorithm that combines two of the existing techniques to deal with data heterogeneity in FL setups.
3. The paper presents exhaustive experimental results to demonstrate the improvements attained by FedDr+
4. The paper also presents results on personalized FL.
5. Ablation study on the newly introduced hyper-parameter i.e., $\beta$ is provided.
6. The paper is well written and easy to follow. The paper address an important concern in Federated Learning i.e., data heterogeneity.

Weaknesses/questions:
1. Contribution 1 indicates that "We find that dot-regression loss is not easily compatible with FL". However, column 2 and 9 of Table 2 shows that dot regression performs better than FedAvg which uses cross-entropy loss. Can you please explain why dot-regression loss is not easily compatible with FL?
2. Will FedDr+ or dot-regression loss outperform  FedAvg in the FL settings with IID data?
3. ETF initialization forces a prior distribution on the class similarities/dissimilarities i.e., all the classes are equally distinct from each other. This prior distribution can be very distinct from the original distribution and can result in poor model performance. How does the algorithm deal with this issue?

---

> ### Author Response · Authors · 2025-02-04
> **Response to Reviewer TUZs (1/3)**
>
> We sincerely appreciate your detailed feedback and constructive suggestions. Below, we provide responses to each of your concerns.
>
> ---
>
> ### Q1) Clarification Response
>
> **A1)** The statement *"dot-regression loss is not easily compatible with FL"* in our paper specifically refers to the **classifier-freezing setup**, where we compared cross-entropy (CE) loss and dot-regression loss. As you correctly noted, when compared to FedAvg without classifier freezing, dot-regression demonstrates better performance.
>
> To address this and improve clarity, we have revised the **Contribution section in the Introduction (page 2, first bullet point)** as follows:
>
>
> > *"In high-heterogeneity FL settings, we observe a trade-off in the classifier-freezing setup: dot-regression loss improves local alignment with observed classes but leads to lower global model performance compared to
> CE loss, due to a significant loss of information on unseen classes, which is critical for the global model."*
>
> This revision clarifies that the trade-off is specific to the classifier-freezing context and aligns with your observation regarding FedAvg. We hope this update addresses your concern. Please let us know if further clarification is needed.
>
>
> ### Q2) IID Data Performance
>
> **A2)** To address the question regarding the performance of FedDr+ or dot-regression loss in Federated Learning (FL) settings with IID data, we conducted experiments on **CIFAR-100 with 100 clients**, distributing data **IID** and ensuring a fair number of samples per client. We evaluated FedAvg, FedBABU, Dot-regression, and FedDr+ across **5 seeds**, calculating the mean and standard deviation of the global model accuracy for each algorithm.
>
> The results are as follows:
>
> | Algorithm          | Accuracy (mean ± std)       |
> |--------------------|--------------------------------------|
> | **FedAvg**         | 47.19 ± 1.06          |
> | **FedBABU**        | 45.18 ± 0.61          |
> | **Dot-regression** | **51.48 ± 0.99**          |
> | **FedDr+**         | **51.10 ± 0.61**          |
>
> From the table, it is evident that **Dot-regression** and **FedDr+** achieve the highest performance, significantly outperforming both FedAvg and FedBABU. The performance of Dot-regression and FedDr+ is nearly identical under IID settings.
>
> This similarity arises because, in the IID scenario, there are no **unobserved classes** across clients. As a result, the feature distillation mechanism in FedDr+, which is specifically designed to mitigate forgetting on unobserved classes, does not provide additional benefits. Instead, both Dot-regression and FedDr+ excel in improving local alignment across all classes, fully achieving the global model's objective of enhancing local alignment for all clients.
>
>
> We have included these results and the corresponding analysis in **Appendix E: Additional Experiment Results**. We hope this explanation clarifies the performance under IID settings.

---

> ### Author Response · Authors · 2025-02-04
> **Response to Reviewer TUZs (2/3)**
>
> ### Q3) Conducting Experiments on Multiple Seeds
>
> **A3)** Thank you for pointing out the importance of conducting experiments on multiple seeds and reporting the mean and standard deviation, as FL with non-IID data often exhibits high variance in performance.
>
> To address this, we have conducted additional experiments using **CIFAR-100 with 100 clients**, running over **5 random seeds** specifically for **Personalized Federated Learning (PFL)** under the following settings:
> - **Shard-based non-IID (s):** Data is split into 10, 20, or 100 shards per client s ∈ {10, 20, 100}.
>
> - **LDA-based non-IID (α):** Dirichlet allocation with concentration parameters α ∈ {0.05, 0.1, 0.3}.
>
> These experiments were performed on **CIFAR-100** with **MobileNet** as the backbone model. The results, reported as **mean ± std**, represent the average personalized accuracy across all clients, computed over five seeds. The table below summarizes the results:
>
> | Algorithm                         | S=10                | S=20                | S=100               | α=0.05            | α=0.1             | α=0.3             |
> |-----------------------------------|---------------------|---------------------|---------------------|-------------------|-------------------|-------------------|
> | Local only (L_CE)                | 58.42 ± 0.22       | 42.37 ± 0.29       | 19.02 ± 0.34       | 55.71 ± 0.19      | 44.02 ± 0.39      | 27.98 ± 0.08      |
> | Local only (L_CE + ETF)          | 58.05 ± 0.25       | 41.72 ± 0.26       | 19.06 ± 0.18       | 55.41 ± 0.21      | 43.56 ± 0.31      | 27.69 ± 0.17      |
> | Local only (L_DR)                | 61.05 ± 0.37       | 44.28 ± 0.21       | 21.06 ± 0.21       | 58.56 ± 0.14      | 47.05 ± 0.13      | 31.16 ± 0.15      |
> | FedPer                            | 70.62 ± 0.71       | 55.65 ± 1.35       | 25.57 ± 0.59       | 63.35 ± 1.96      | 51.90 ± 2.13      | 35.84 ± 2.16      |
> | Per-FedAvg                        | 31.71 ± 1.08       | 38.64 ± 0.40       | 45.71 ± 0.81       | 28.85 ± 0.27      | 36.00 ± 0.42      | 42.41 ± 0.32      |
> | FedRep                            | 62.59 ± 0.30       | 51.18 ± 1.00       | 26.51 ± 0.27       | 57.73 ± 0.41      | 49.59 ± 0.40      | 36.22 ± 0.86      |
> | Ditto                             | 38.39 ± 0.54       | 42.16 ± 1.14       | 44.04 ± 0.81       | 34.86 ± 1.18      | 38.67 ± 1.30      | 42.05 ± 0.58      |
> | **FedAvg-FT**                     | 70.20 ± 0.54       | 56.26 ± 0.51       | 48.67 ± 0.99       | 61.08 ± 1.86      | 56.34 ± 1.18      | 49.74 ± 1.08      |
> | **FedBABU-FT**                    | 80.73 ± 0.65       | 71.02 ± 0.34       | 51.70 ± 0.21       | 76.12 ± 0.55      | 69.94 ± 0.34      | 57.40 ± 1.50      |
> | **SphereFed-FT**                  | 81.34 ± 0.64       | 72.22 ± 0.56       | 56.58 ± 0.89       | 74.49 ± 0.86      | 69.39 ± 1.04      | 59.51 ± 1.03      |
> | **FedETF-FT**                     | 53.32 ± 0.60       | 53.05 ± 0.49       | 49.74 ± 0.85       | 52.31 ± 0.40      | 53.70 ± 0.35      | 50.80 ± 0.65      |
> | **FedGELA-FT**                    | 75.75 ± 0.57       | 68.96 ± 0.37       | 52.23 ± 0.59       | 58.26 ± 5.78      | 60.12 ± 0.71      | 53.09 ± 0.82      |
> | **FedDr+FT (ours)**               | **83.08 ± 0.27**   | **74.80 ± 0.66**   | **56.56 ± 1.04**   | **78.40 ± 0.40**  | **73.23 ± 0.89**  | **62.22 ± 0.86**  |
>
> For **Global Federated Learning (GFL)**, the results already include mean and standard deviation values over three seeds. Therefore, this additional analysis focuses solely on **PFL**.
>
> As shown in the table, **FedDr+FT** consistently achieves the best performance across all settings, demonstrating its robustness and effectiveness in non-IID environments.
>
> We have updated the manuscript to include this analysis, replacing **Table 5** in the main text with the above table to provide comprehensive results.

---

> ### Author Response · Authors · 2025-02-04
> **Response to Reviewer TUZs (3/3)**
>
> ### Q4) **Response to ETF Prior Distribution Concern**
> **A4)** While real-world data distributions may not inherently follow the **ETF prior**, freezing the classifier as an **ETF preserves class separation** under class imbalance in **centralized learning** and extends these benefits to **FL by stabilizing classifier consistency across clients**.
>
> ### **ETF in Centralized Learning**
> Neural Collapse [1] naturally occurs in **class-balanced centralized learning**, where:
> - Classifier vectors form an ETF, ensuring maximal class separation.
> - Feature prototypes align with classifier vectors, enhancing discriminability and compactness.
>
> However, **Neural Collapse does not naturally emerge under class imbalance [2]**. [2] theoretically proves that freezing the classifier as an ETF preserves Neural Collapse even in class-imbalanced settings, providing a principled approach to maintaining class separability. Additionally, **[3] empirically supports that a well-separated classifier improves performance**, reinforcing ETF’s validity.
>
> ### **Extending ETF Benefits to FL**
> FL introduces data heterogeneity, leading to client-specific class distributions and classifier inconsistencies across local models. By **freezing the classifier as an ETF**, all clients align to a globally consistent classifier with maximally separated class vectors, preventing divergence and stabilizing learning [4].
>
> Thus, despite differences from real-world data distributions, **ETF provides structured class separation beneficial for both centralized and FL settings**.
>
> ---
>
> ### Q5) Additional Details: Pseudo Code and Limitations
>
> **A5)** We have added a pseudo code for **FedDr+** and **FedDr+FT**  in **Appendix A** to provide an implementation guide.
>
> Regarding the limitations, we have created a dedicated **Limitation** section in the main text to address the following points:
>
> - While our method uses dot-regression to optimize local alignment for observed data, this is just **one approach**. Other strategies, such as directly maximizing local alignment, could be explored and generalized in future work.
> - We find that dot-regression is less effective in preserving local alignment for unobserved classes. While our proposed FedDr+ addresses this limitation empirically, developing a **theoretical justification** for improving unobserved class preservation remains an important direction for future research.
>
> These points highlight opportunities to further generalize and enhance the proposed method in Federated Learning.
>
> ---
> ### **Manuscript Revisions**
>
>  We have revised the manuscript as follows:
>  - **Introduction (page 2, first bullet point)**: Clarified that dot-regression improves local alignment for observed classes compared to CE but degrades alignment for unobserved classes.
> - **Appendix E**: Added IID performance results for **FedDr+ vs. Dot-regression vs. FedBABU vs. FedAvg**.
> - **Appendix A**: Added a **pseudo-code implementation of FedDr+**.
> - **Limitation Section (Main Text)**: Discussed **generalization challenges and the need for further theoretical analysis of FedDr+**.
> ---
> ### **References**
>
> [1] Prevalence of neural collapse during the terminal phase of deep learning training, PNAS 2020
>
> [2] Inducing Neural Collapse in Imbalanced Learning: Do We Really Need a Learnable Classifier at the End of Deep Neural Network?, NeurIPS 2022
>
> [3] FedBABU: Towards Enhanced Representation for Federated Image Classification, ICLR 2022
>
> [4] No Fear of Classifier Biases: Neural Collapse Inspired Federated Learning with Synthetic and Fixed Classifier, ICCV 2023

---

### Review · Reviewer_f4TB · 2025-01-21

**Summary Of Contributions:**

The paper introduces FedDr+, an approach designed to improve performance in both global and personalized federated learning (FL) scenarios with heterogeneous client data. The method focuses on enhancing local alignment while preserving the representation of unseen class samples. The authors adopt a dot-regression loss to strengthen local alignment, while they also identify a drawback of this approach: a significant loss of information related to unseen classes. To maintain global knowledge, FedDr+ incorporates a feature distillation mechanism during local model training. This ensures the model does not overly prioritize local alignment at the expense of broader generalization. The paper includes several experiments to validate the effectiveness of FedDr+.

**Audience:**

Yes

**Claims And Evidence:**

Yes

**Requested Changes:**

Please see above

**Strengths And Weaknesses:**

Strengths

* The paper is well-written and easy to follow.
* Improving performance in both global and personalized FL with heterogeneous data is an important open problem, and the authors provide a sound method.
* Each component of FedDr+ is introduced with clear motivation and supported by sufficient experimental evidence.
* The evaluations are comprehensive and well-executed.

Weaknesses and Questions

* Beyond the intuitive explanation, is there any theoretical justification for why CE performs better than DR on unobserved data?
* The number of clients and training rounds is small. What happens if training rounds exceed 1k and the number of clients exceeds 1k?
* Does the observed/unobserved data class change in each training round?
* What happens if some data classes appear only at the beginning and then never reappear? Does FedDr+ still retain those features?

---

> ### Author Response · Authors · 2025-02-04
> **Response to Reviewer f4TB (1/2)**
>
> We sincerely appreciate your detailed feedback and constructive suggestions. Below, we provide responses to each of your concerns.
>
> ---
>
> ### **Q1) Scaling to Larger Numbers of Clients and Training Rounds**
>
>  **A1)** We conducted experiments on **CIFAR-100 (s=10)** with **1,000 communication rounds**, increasing the number of clients to **100, 200, 500, and 1,000**. All algorithms used previously **grid-searched optimal hyperparameters**, and results are averaged over **three independent seeds**.
>
> | **Algorithm** | **N=100**           | **N=200**           | **N=500**           | **N=1,000**         |
> |--------------|---------------------|---------------------|---------------------|---------------------|
> | **FedAvg**   | 50.50 ± 0.57        | 42.51 ± 1.47        | 33.02 ± 0.74        | 26.63 ± 1.31        |
> | **FedBABU**  | 58.19 ± 1.07        | 48.75 ± 1.99        | 37.40 ± 0.41        | 25.10 ± 1.08        |
> | **FedDr+**   | **64.21 ± 1.24**    | **59.78 ± 0.71**    | **43.27 ± 0.31**    | **28.99 ± 0.98**    |
>
> These results confirm that **FedDr+ consistently outperforms FedAvg and FedBABU** across all settings, demonstrating **robust scalability** in large-scale FL.
>
> To ensure clarity, we have included these results in **Appendix E** of the revised manuscript.
>
> ---
>
> ### **Q2) Performance in Stochastic Client Data Settings**
>
> **A2)** While our original experiments on **CIFAR-100 (s=10) with 100 clients and **320 communication rounds**** assumed a static client dataset, we conducted additional experiments where each client randomly removed one class from its dataset every 10 FL rounds. As expected, global model accuracy decreased for all methods. However, **FedDr+ consistently outperformed CE  and Dot-regression**, demonstrating its robustness in handling **dynamic class distributions**.
>
> #### **Global Model Accuracy (%) in Static and Stochastic Settings**
>
> | **Algorithm**       | **Static Setting** | **Stochastic Setting** |
> |---------------------|--------------------|------------------------|
> | **CE**        | 46.20              | 43.59                  |
> | **Dot-regression** | 42.52              | 38.13                  |
> | **FedDr+**         | **48.69**          | **44.96**              |
>
> To better understand this robustness, we conducted two additional analyses:
>
> 1. **Local alignment for observed/unobserved classes:**
>    - **FedDr+ maintains superior feature-classifier alignment for both observed and unobserved classes** compared to Dot-regression, consistently outperforming it across all rounds.
>    - During the final convergence phase, **FedDr+ surpasses even CE in unobserved class alignment**, confirming its effectiveness in preserving global knowledge.
>
> 2. **Local alignment for erased classes:**
>    - Even for erased classes (those removed during training), **FedDr+ retains stronger feature-classifier alignment** than Dot-regression.
>    - **During the final convergence phase, FedDr+ also surpasses CE in erased class alignment**, further demonstrating its ability to mitigate forgetting of removed class knowledge.
>
> These results suggest that the **feature distillation mechanism in FedDr+ effectively enhances global knowledge preservation while also enabling effective learning of observed classes, even when class distributions change dynamically**. Additional **figures on local alignment and global accuracy trends** are available in **Appendix E**.

---

> ### Author Response · Authors · 2025-02-04
> **Response to Reviewer f4TB (2/2)**
>
> ### **Q3) Theoretical Justification: Why CE Outperforms Dot-Regression on Unobserved Data**
>
> **A3)** We provide a theoretical justification in **Appendix C: Theoretical Perspective of Dot-Regression (DR)** by analyzing **feature gradients** and the **Neural Tangent Kernel (NTK) regime** to explain why **cross-entropy (CE) loss performs better than dot-regression (DR) on unobserved classes**.
>
> #### **Feature Gradient of Dot-Regression**
> Dot-regression updates features by adding a component **orthogonal** to the current feature vector, rotating it toward the target classifier vector \( v_y \) while increasing feature norm. As alignment improves, the update magnitude decreases, slowing down feature updates near convergence.
>
> #### **NTK Perspective: Why Dot-Regression Fails on Unobserved Classes**
> We analyze why dot-regression struggles with unobserved classes using the **NTK regime**. In this framework, the feature gradient for any input is expressed as a **weighted combination** of feature gradients from the training dataset, where weights are **entirely determined** by the **input** and the **distribution of the initialized model**. The NTK framework provides a well-established approximation for deep models with large width.
>
> For a given **unobserved class** \( c \) in a local client with classifier vector \( v_c \), dot-regression **does not update features toward \( v_c \)** because class \( c \) is absent from local training data. As a result:
>
> - **Feature gradients remain independent of \( v_c \)** since, under the NTK framework, the feature gradient of an unobserved class \( c \) is a weighted sum of feature gradients from the training dataset. Without class \( c \), these gradients lack alignment with \( v_c \), preventing local learning from improving alignment.
> - **Alignment for unobserved classes relies on global aggregation**, where updates from other clients containing class \( c \) indirectly contribute to alignment.
>
>
> This theoretical insight explains why dot-regression alone is insufficient for aligning unobserved classes in local training.
>
>
> In contrast, **CE loss explicitly moves feature gradients in the direction of \( v_c \)**, weighted by the **softmax probability \( p_c \)** and the **NTK weight**. This ensures that even for **unobserved classes**, local training contributes meaningful feature updates. After **global aggregation**, the updated \( p_c \) further refines these updates, allowing **CE to maintain better feature-classifier alignment across all classes**.
>
> This theoretical analysis supports our empirical findings:
> **Dot-regression struggles to maintain alignment for unobserved classes, whereas CE enables continued feature updates, leading to better generalization.**
>
> We have included this discussion in **Appendix C** of the revised manuscript.
>
> ---
>
> ### **Manuscript Revisions**
>
> To ensure clarity, we have revised the manuscript as follows:
> - **Appendix E** now includes results from the **scalability experiments** (1,000 rounds, varying client numbers).
> - **Appendix E** also includes new results for **stochastic client data settings**.
> - **Appendix C** has been updated with a **theoretical discussion on NTK analysis**, further supporting why **CE performs better than Dot-regression**.

---

### Author Response · Authors · 2025-02-04
**General Response**

Dear Reviewers **(@f4TB, @TUZs, @BYb6)**,

We sincerely appreciate your **thorough reviews** and **insightful feedback**. Your recognition of our work’s **importance, clarity, and experimental rigor** motivates us to refine our contributions.

### **Key Strengths Acknowledged by Reviewers**
- **Well-written and easy to follow** (**@f4TB, @TUZs**).
- **Sound method** for **both global and personalized FL** (**@f4TB, @TUZs, @BYb6**).
- **Simple and practical**, ensuring **easy integration** into FL frameworks (**@BYb6**).
- **Well-motivated components** with **strong experimental validation** (**@f4TB, @TUZs, @BYb6**).
- **Comprehensive evaluations**, including **ablation studies and hyperparameter analysis** (**@TUZs**).


To further improve the manuscript and address the **reviewers' concerns**, we have **revised the paper as follows**:

### **Manuscript Revisions**

- **Introduction:**
  - Clarified **local alignment** and **global knowledge preservation** (**@BYb6**).
  - Explained how **FedDr+FT extends FedDr+ to PFL** (**@BYb6**).
  - Revised **Contribution section** to clarify the statement on **dot-regression compatibility in FL** (**@TUZs**).

- **Experiments and Results:**
  - Updated **Table 5** to include **results over multiple seeds, reporting mean and standard deviation** (**@TUZs**).

- **Limitation Section:**
  - Added discussion on **generalization challenges** and **future directions** (**@TUZs**).

- **Appendix A:**
  - Added **pseudocode** for **FedDr+ and FedDr+FT** (**@TUZs**).

- **Appendix C:**
  - Provided **NTK-based theoretical justification** for **dot-regression on unobserved classes** (**@f4TB**).

- **Appendix E:**
  - Added **IID dataset results** (**@TUZs**).
  - Added **stochastic client data results** (**@f4TB**).
  - Included **scalability experiments**  (**@f4TB**).


These revisions enhance the **clarity, theoretical foundation, and practical insights** of our paper. We sincerely appreciate the reviewers' valuable time and effort in improving our work and welcome any further feedback.


Sincerely,
-Authors-

---

### Decision · Action_Editor_67Ps · 2025-02-24

**Recommendation:** Accept as is

**Comment:**

The authors experimentally validate their federated learning algorithm. In particular the reviewers are all convinced, especially after response period, that there are good performance improvements in the case of heterogeneous data setting of federated learning. The authors have shown that the improvement persists both in the global sense, and also in a personalized (locally measured) sense. While the theoretical justification is lacking, the method seems to be promising to federated learning practitioners.

**Audience:**

Practice of federated learning has garnered significant attention in the past few years - undoubtably this will be of interest to such practitioners.

**Claims And Evidence:**

The paper a new federated learning method that combines the a new loss function, called the dot-regression loss with a feature distillation mechanism, to mitigate the effects of client drift. Thorough empirical evidences were provided to support the proposed method.